# Evaluation and Exploration of Machine Learning and Convolutional Neural Network Classifiers in Detection of Lung Cancer from Microarray Gene—A Paradigm Shift

**DOI:** 10.3390/bioengineering10080933

**Published:** 2023-08-06

**Authors:** Karthika M S, Harikumar Rajaguru, Ajin R. Nair

**Affiliations:** 1Department of Information Technology, Bannari Amman Institute of Technology, Sathyamangalam 638401, India; mskarthika315@gmail.com; 2Department of Electronics and Communication Engineering, Bannari Amman Institute of Technology, Sathyamangalam 638401, India; rn.ajin@gmail.com

**Keywords:** lung cancer classification, dimensionality reduction, feature selection techniques, Short Term Fourier Transform, Particle Swarm Optimization, Harmonic Search, Nonlinear Regression, Gaussian Mixture Model, Softmax Discriminant, Naive Bayes, Support Vector Machine, Convolutional Neural Network for Lung Cancer, microarray gene expression dataset

## Abstract

Microarray gene expression-based detection and classification of medical conditions have been prominent in research studies over the past few decades. However, extracting relevant data from the high-volume microarray gene expression with inherent nonlinearity and inseparable noise components raises significant challenges during data classification and disease detection. The dataset used for the research is the Lung Harvard 2 Dataset (LH2) which consists of 150 Adenocarcinoma subjects and 31 Mesothelioma subjects. The paper proposes a two-level strategy involving feature extraction and selection methods before the classification step. The feature extraction step utilizes Short Term Fourier Transform (STFT), and the feature selection step employs Particle Swarm Optimization (PSO) and Harmonic Search (HS) metaheuristic methods. The classifiers employed are Nonlinear Regression, Gaussian Mixture Model, Softmax Discriminant, Naive Bayes, SVM (Linear), SVM (Polynomial), and SVM (RBF). The two-level extracted relevant features are compared with raw data classification results, including Convolutional Neural Network (CNN) methodology. Among the methods, STFT with PSO feature selection and SVM (RBF) classifier produced the highest accuracy of 94.47%.

## 1. Introduction

Lung cancer remains one of the most significant health concerns worldwide, with high morbidity and mortality rates. It is primarily associated with exposure to tobacco smoke, either through active or passive smoke. However, as mentioned in Dubin et.al. [1], non-smokers may also develop lung cancer due to environmental pollutants, occupational hazards, genetic predisposition, and underlying lung diseases. The risk increases with prolonged exposure to carcinogens, such as asbestos, radon, and certain industrial chemicals. In recent years, there has been increasing emphasis on early detection and intervention of lung cancer. At early stages, the tumor is localized; hence the treatment options are more effective, and the chances of successful intervention and cure are significantly high. Unfortunately, as sermonized in Selman et al. [2], lung cancer often remains asymptomatic or presents nonspecific symptoms until it reaches an advanced stage, making early detection challenging.

Various detection methods have been employed to identify lung cancer early. Infante et al. [3], utilized chest X-rays and Computed Tomography (CT) scan imaging techniques to detect suspicious lung nodules, masses, or other abnormalities that may indicate the presence of lung cancer. The Sputum Cytology method used in Thunnissenet al. [4] analyzes the sputum sample for the presence of cancer cells. There are also procedures such as Bronchoscopy studied in Andolfiet al. [5] that allow the collection of lung tissue samples and visualization of the airways using a thin, flexible tube and camera. All of the above methods or procedures are conducted in the primary stages after the spread of lung cancer cells in the individual. However, Zhu et al. [6] used the method of microarray-based gene expression analysis to detect lung cancer at its earliest stages, facilitating timely intervention and improving patient outcomes. The microarray analysis compares healthy lung tissue’s gene expression profiles with cancerous tissue. The analysis aids the researchers in identifying distinct patterns and signatures associated with different lung cancer types.

### 1.1. Review of Previous Works

As mentioned, microarray based gene expression analysis is widely used for the detection and classification of various health conditions. A microarray is a tool used in molecular biology to study gene expression. As described by Churchill et al. [7], microarray is a collection of microscopic spots containing DNA molecules representing individual genes. These spots are arranged in a grid pattern on a small glass slide or a silicon chip. Microarray gene expression analysis can investigate changes in gene expression patterns in cancer cells compared to normal cells, as sermonized in Ross et al. [8]. By simultaneously measuring the expression levels of thousands of genes, microarrays can provide a comprehensive view of the changes in cancer cells. Also, as reported by Reis et al. [9], microarray gene expression analysis can be utilized to classify different types of cancer, which can help and guide treatment decisions at early stages.

As indicated in Lapointe et al. [10], different types of cancer have distinct gene expression profiles, which reflect the underlying molecular and genetic changes that drive the disease. In Dwivedi et. al. [11], researchers have used microarray analysis to identify gene expression patterns that distinguish between different types of leukemia, such as acute lymphoblastic leukemia (ALL) and acute myeloid leukemia (AML). Similarly, Rody et al. [12] used microarray analysis to distinguish between different breast cancer subtypes, such as estrogen receptor-positive and HER2-positive breast cancer. In Sanchez et al. [13], microarray gene expression analysis is employed to predict the chances of tumor and lung cancer. Kerkentzes et al. [14] classified Adenocarcinoma (Adeno) and Mesothelioma (Meso) lung cancer from microarray genes. Adeno and Meso are considered serious cancer conditions for diverse reasons mentioned in Wagner et al. [15]. Adeno can exhibit rapid tumor growth and potentially spread to other body parts, including distant organs. Meso is extremely aggressive, making the treatment difficult when diagnosed at advanced stages. Both Adeno and Meso have limited treatment options compared to other forms of lung cancer. Although surgery, chemotherapy, and radiation therapy can be used, these treatments may not always be curative, especially if the cancer has spread all over the body.

Further, these cancer conditions develop resistance to standard cancer treatments due to genetic mutations in tumor cells, limiting the effectiveness of targeted therapies. Above all, the significance of this research investigation is the poor prognosis of Adeno and Meso conditions compared to other forms of lung cancer. The outcome of this research will help in timely detection, early intervention, and personalized treatment approaches to improve the conditions of cancer-diagnosed individuals. Microarray gene expression analysis-based lung cancer detection offers several advantages over traditional histopathology diagnostic methods, as sermonized in Weigelt et al. [16]. Unlike histopathology, gene expression analysis provides molecular insights into the underlying biology of lung cancer, which helps to understand lung cancer subtypes, disease progression, and treatment responses. The molecular information goes beyond the structural or morphological features captured by imaging or histopathology, providing a deeper understanding of the disease. Also, traditional diagnostic methods may only detect lung cancer when it has already reached a more advanced stage. Microarray gene expression analysis also brings up potential biomarkers with higher sensitivity and specificity than traditional diagnostic methods. All of these benefits motivate us to use microarray gene expression analysis for early diagnosis of lung cancer.

### 1.2. Review of Feature Extraction Techniques

Feature extraction techniques are crucial in acquiring relevant features from microarray gene data. As described in Hiraet al. [17], these methods aim to reduce the dimensionality of the data while retaining important information for subsequent analysis. The high dimensional data are reduced so that it can explain most of the variance in the dataset. The reduced data also contains the essential patterns and relationships between genes, entitling a more manageable and meaningful representation of the data. Feature extraction methods are advantageous in removing noise by extracting the underlying signal or patterns in the data by focusing on the most significant features.

In this paper, we employ Short-Time Fourier Transform (STFT) for analyzing and expressing the microarray gene expression data. The STFT provides a time-frequency representation of the signal, which can capture changes in gene expression over time. The STFT provides information about the temporal localization of signal events, allowing researchers to identify specific time points or intervals where gene expression changes occur as mentioned in Qi et al. [18]. This representation can also reveal temporal patterns and relationships between genes, allowing for the identification of dynamic gene expression changes associated with specific conditions or biological processes. The STFT can also extract frequency-domain features from gene expression data. These features provide additional information about the distribution or characteristics of gene expression patterns that help to improve classification accuracy. Also, STFT identifies specific frequency bands representing relevant genes associated with biological processes and disease conditions.

### 1.3. Review of Feature Selection Techniques

After feature extraction, feature selection methods further improve lung cancer classification from microarray gene expression data. As described in Abdelwahab et al. [19], these methods help identify a subset of relevant genes that are informative for distinguishing between different lung cancer types. In this paper, we employ metaheuristic feature selection methods namely Particle Swarm Optimization (PSO) and Harmonic Search (HS) for selecting this subset of genes most discriminatory for lung cancer classification. As elaborated in Shukla et al. [20], by focusing on the most discriminative genes, these metaheuristic methods can reduce the risk of overfitting and improve the generalization capability of the classifier, leading to better performance. Also, this is extremely useful for classifying the LH2 dataset, as the number of samples is limited. Metaheuristic feature selection methods can also handle correlations and dependencies between genes to remove redundant genes while maximizing their relevance to lung cancer classification. Overall, the metaheuristic feature selection methods identify a biologically relevant gene subset and can give insights into the underlying biological processes associated with lung cancer. But, the choice of metaheuristic algorithm for the feature selection method plays a vital role in selecting the relevant genes from the dataset. So it is important to evaluate different metaheuristic methods and consider their performance, computational complexity, and suitability for the specific problem.

### 1.4. Review of CNN Methodology

Apart from machine learning classifiers, Convolutional Neural Networks (CNNs) may make a significant difference in the classification of lung cancer from microarray gene expression data. Without manual feature extraction and selection methods, CNNs can automatically learn relevant features from raw input data. In the context of gene expression data, CNNs can extract meaningful patterns and relationships between genes. This methodology allow CNNs to capture complex interactions that may be difficult to discern manually. For gene expression data, genes can be considered spatial entities, and their expressions across samples can be treated as spatial patterns. Here, CNNs excel at learning hierarchical representations, starting from low-level gene expression features and gradually building up to higher-level features that capture more abstract characteristics. Also, CNNs can identify gene expression patterns relevant to lung cancer, even if they occur in different regions or orders within the microarray. CNNs employ parameter optimization and deduce conclusions better from limited training data like microarray gene expression datasets. Further, CNNs are robust to inherent noise, variability, and batch effects variations. CNNs can learn to identify important patterns despite these variations, making them suitable for analyzing microarray gene expression datasets. Also, CNNs models inherently form nonlinear relationships between features crucial for capturing the complex and nonlinear interactions within gene expression data. So we also use experiments on employing CNN methodology to classify lung cancer from microarray gene expression data. The overall methodologies for dimensionality reduction and classification of microarray gene expression data for lung cancer employed in this paper are shown in Figure 1.

Despite the numerous benefits of using CNNs for classifying lung cancer from microarray gene expression data, they also have certain limitations. Even though CNNs are good at learning features, it is often challenging to interpret the physical significance behind the prediction and classification of gene expression patterns from the learned features due to their black-box nature. Further, CNNs are usually prone to overfitting due to the limited dataset availability; hence, it is important to carefully tune and select the parameters in the CNN model. The feature extraction and selection methods can solve the overfitting problem to a certain extent in CNN models. However, these stages also add computational complexity to CNNs’ inherent complexity. For the microarray gene expression dataset used in this paper (Lung Harvard 2 Dataset), the imbalanced class distributions in the lung cancer dataset can pose challenges. Due to class imbalance, CNNs may tend to bias towards the Adeno class, leading to lower accuracy for the Meso class. The performance of CNN can be improved by carefully choosing tuning parameters like learning rate, network architecture, and regularization strength. Also, suitable classifiers are integrated into the softmax layer to improve classification performance in CNN.

### 1.5. Description of the Dataset

In this research, we have utilized the Lung Harvard 2 (LH2) medical imaging dataset [21]. This dataset contains translated microarray data of patient samples affected by Adenocarcinoma (Adeno) and Mesothelioma (Meso). The gene expression ratio technique is the translation technique used in the dataset for a simpler representation. The technique can accurately differentiate between genetically disparate and normal tissues based on thresholds. The dataset contains a total of 150 Adeno and 31 Meso tissue samples. The 12533x 1 genes characterize each of the subject. The final row of the dataset is used for labelling Adenocarcinoma and malignant Mesothelioma samples. This microarray gene expression dataset is widely used for early and authentic diagnosis of lung cancer classification.

### 1.6. Relevance of This Research

As mentioned earlier, lung cancer detection from microarray gene data helps to diagnose cancer at an earlier stage, which is crucial as the treatment options are limited in advanced stages. Thus, early detection can significantly improve patient outcomes and survival rates. In this paper, we perform the lung cancer classification of Adeno and Meso cancer from the LH2 dataset. The dataset is of high volume and contains inherent nonlinearity and class imbalance. Therefore, it is important to adopt suitable preprocessing and classification methods to enhance the classification performance. Thus, learning on microarray gene data analysis enables the identification of potential biomarkers associated with lung cancer to help in diagnosis, prognosis, and monitoring treatment response.

## 2. Methodology

In this research, a two-level strategy is adopted to improve the overall efficiency of the classifiers. First, feature extraction is performed using STFT to solve the curse of dimensionality problem. The dimension of the dataset is reduced to leverage the computational overhead of the classifier. Further, the second step involves selecting features from PSO and HS metaheuristic algorithms. The feature selection step further improves the classification methodology by selecting relevant features and patterns and reducing the risk of overfitting and underfitting.

### 2.1. Feature Extraction Using STFT

The Short-Time Fourier Transform (STFT) is a frequency domain analysis of the information over a short period. The STFT, when applied to microarray gene data, can reduce the data dimension by extracting relevant and useful features. Gupta et al. [22] have performed QRS Complex Detection Using STFT. The authors have identified that STFT is useful in providing a time-frequency representation of the data, which helps researchers investigate how gene expression levels change over time and across different frequency components. Identifying specific time intervals where certain frequency components are prominent is possible with STFT, giving a localized representation of frequency content over time. For microarray gene expression analysis, this feature of STFT is useful in identifying genes that exhibit temporal patterns active during specific time intervals. Moreover, STFT reduces dimensionality by extracting the important genes or gene clusters associated with specific frequency components. The dimensionality of the microarray gene expression dataset is reduced to 2049 × 150 and 2049 × 31 for Adeno and Meso subjects, respectively. For performing STFT, the Blackman window that minimizes spectral leakage is used as the windowing function. Thus, STFT can provide biological insights by finding the frequency patterns and fundamental relationships in the dataset.
(1)$$X\left(m,w\right)=\sum_{n=-\infty }^{\infty }x\left[n\right]\;w\left[n-m\right]e^{-jwn}$$

Here, *x*[*n*] represents the input data having length *N* with *n* = 0, 1, 2, … *N*−1. The STFT window is represented by *w*[*n*] having length *M* with *m* = 0, 1, 2, … M−1, j = √ (−1) denotes the complex number. The Blackman window is given by the expression
(2)$$w\left[n\right]=0.42-0.5\;\cos\left(\frac{2\pi n}{M}\right)+0.08\;\cos\left(\frac{4\pi n}{M}\right),\;\;0\;\leq n\;\leq N-1$$

Next section is an analysis on the various statistical parameters associated with STFT dimensionality reduction.

### 2.2. Statistical Analysis on STFT

After the dimensionality reduction methods on microarray genes, the resultant outputs are analyzed by the statistical parameters such as mean, variance, skewness, kurtosis, Pearson correlation coefficient (PCC), f-test, *t*-test, *p*-value, and Canonical correlation Analysis (CCA) to identify whether the outcomes are representing the underlying microarray genes properties in the reduced subspace. Table 1 shows the statistical features analysis for the STFT Dimensionally Reduced (DR) Adeno Carcinoma and Meso Cancer Cases of microarray genes. As mentioned in Table 1, the STFT features show higher mean, variance and kurtosis values among the two classes. Skewness values are observed to be similar with positive skewness. PCC values indicate the high correlation within the class of the attained outputs. The permutation and sample entropy for the Adeno and Meso cancer cases are similar. The f-test, *t*-test and *p*-value exhibit no significant nature of the dimensionally reduced outputs of the microarray gene. The statistical f-test and *t*-test are performed for the Null Hypothesis. The *t*-test value is low for the Meso cancer class but does not indicate significance as the *p*-value is higher than 0.01. This statistical analysis indicates that the values are associated with non-Gaussian and nonlinear ones. The histogram, Normal probability plots, and Scatter plots of DR techniques output further examine the same. CCA visualizes the correlation of DR methods outcomes among the Adeno Carcinoma and Meso Cancer Cases. The low CCA values in Table 1 indicate that the DR outcomes are less correlated among the two cancer classes.

Figure 2 explores the Scatter plot for the STFT-based Dimensionality Reduction Method in Meso and Adeno Carcinoma Cancer Classes. 

After applying STFT-DR, the Meso and Adeno Carcinoma Cancer dataset seems clustered and overlapped in the scatter plot as most data are amalgamated into a single group. The data after STFT-DR also does not show any signs of Gaussian distribution. Therefore, the scatter plot emphasizes the need for the feature selection process that enhances the classification accuracy of the classifiers. 

### 2.3. Feature Selection Using Metaheuristic Algorithms

As described before, the dimensionality reduction step reduces computational complexity and removes noise in the data. The feature selection step further reduces the number of features after the dimensionality reduction step making subsequent computations faster and more efficient. Also, the feature selection step acts as one more noise filter stage that captures only the relevant features that contribute to the underlying structure of the data. The feature selection step reveals the relevant features from the dimensionally reduced data. This subset of the significant features is the most informative one that holds the gene patterns and functionality to develop a simpler model. The feature selection using metaheuristic algorithms helps to overcome the problems of computationally expensive and noise-hidden datasets by employing efficient search strategies. The algorithms, namely PSO (Particle Swarm Optimization) and Harmony Search (HS), are known for effectively handling high dimensional and complex datasets like microarray gene datasets. Metaheuristic algorithms reduce the microarray gene expression datasets considering the original number of samples and the required number of genes. The reduced dataset contains complex relationships between genes and their associations with biological processes. Metaheuristic algorithms evaluate the collective behavior of the population to arrive at optimal feature subsets. The optimal feature subset assists the subsequent stages, like disease classification.

PSO is an optimization algorithm inspired by the food search and social behavior of a flock of birds. When applied in the feature selection step, the PSO explores and refines the population by moving through the entire search space, balancing exploration and exploitation. Over the iterations, PSO gradually refines the solution space towards the global optimum. The implementation of PSO is simple and efficient as it involves fewer control parameters than other metaheuristic algorithms. Therefore, PSO converges relatively fast, making it computationally efficient for feature selection from large datasets like microarray gene expression datasets. The HS, on the other hand, is inspired by the process of improvisation in music. The generation of harmonious musical notes represents potential solutions. The HS employs a unique approach to enhance the search space exploration by dodging local optima. The HS incorporates adaptive parameters that dynamically change during the search process. This behavior is useful because HS can easily adapt to the characteristics of the microarray gene expression dataset. The HS is also robust to noise and nonlinearities in data, making it a suitable method for feature selection from microarray datasets. So, based on the peculiar characteristics possessed by PSO and HS, these algorithms are used as the feature selection step in this research to understand gene interactions, capture complex relationships, balance exploration and exploitation, and thus identify the most relevant genes. Both objective and fitness functions in the PSO problem are to minimize the Mean Square Error (MSE) of the training process. The relevant features selected after the metaheuristic method will further reduce the dimensionality of the STFT feature extracted values to 2049 × 30 and 2049 × 6 for Adeno and Meso cancer subjects, respectively.

The normal probability plot for the STFT DR method followed by the PSO feature Selection technique for Adeno Carcinoma Cancer Classes is shown in Figure 3. It is also observed from the Figure 3 that the PSO feature selection turned around the STFT features into a clustered one with a sigmoid structure having a mean value of 2.56. The normal plot also implies that the PSO features are non-Gaussian and exhibit minimum segregation. The data 1 through data 5 represents the reference line. The data 6 through data 10 represents the upper bound. The data 11 through data 15 are the feature selected PSO values for the Adeno class.

The normal probability plot for the STFT DR method followed by the PSO feature Selection technique for Meso Carcinoma Cancer Classes is shown in Figure 4. It is also observed from Figure 4 that the PSO feature selection clustered the STFT features around its mean value of 2.47. The normal plot also looks like a sigmoid structure. The normal plot also implies that the PSO features are non-Gaussian and exhibit minimum segregation. The data 1 through data 5 represents the reference line. The data 6 through data 10 represents the upper bound. The data 11 through data 15 are the feature selected PSO values for the Meso class.

Figure 5 displays the histogram of the STFT DR method followed by Harmonic Search Feature selection techniques for Adeno Carcinoma Cancer classes. The histogram also depicts harmonic search features with outliers, wide gaps, down trends and non-Gaussian nature. The STFT dimensionality reduced Harmonic Search feature selection data for four Adeno Carcinoma class patients is denoted by x(:,1), x(:,2), x(:,3), and x(:,4).

Figure 6 exhibits the histogram of the STFT DR method followed by Harmonic Search Feature selection techniques for Meso Carcinoma Cancer classes. The histogram also displays harmonic search features with outliers, wide gaps, skewed, and non-Gaussian nature. The STFT dimensionality reduced Harmonic Search feature selection data for four Meso Carcinoma class patients is denoted by x(:,1), x(:,2), x(:,3),and x(:,4).

Table 2 shows the analysis of the Friedman test in Feature selection methods on STFT Data. It is observed from Table 2 that the PSO and Harmonic Search for the both cancer classes and the feature selection methods depicts the non-significance *p*-values.

The Friedman test is one type of statistical test which tests the stationarity and two-way analysis of variance. It is an extension of the sign test for matched pairs and is used when the data arise from more than two related samples. It is applied to determine whether any dependent variable significantly affects the feature selection method. It is employed in this research as a measuring tool to compare the performance of feature selection algorithms on the same dataset and assess their relative performance. Thus, the Friedman test helps to identify the most effective method among the evaluated set. Since it is a non-parametric test, it is particularly useful when data have heterogeneity or are not following a normal distribution. The Friedman test also provides a statistical measure to determine whether the observed differences in performance are significant. Overall, the test is useful in deciding the relative efficacy of the feature selection techniques. At first, suitable parameters attained from the Friedman test are decided, and performance is evaluated multiple times to account for variability. After that, methods are ranked such that the best method receives a rank of 1, the second best receives a rank of 2, and so on. Consecutively average rank is also calculated. Finally, the Friedman statistic is computed using the following expression.
(3)$$Friedman\;statistics=\left[\frac{12}{n\;k(k+1)}\sum_{i=1}^{k}{R_{i}}^{2}-3n(k+1)\right]$$

Here, k represents the number of feature selection methods, n denotes the number of repetitions, and R indicates the average rank of each method. This research measures the efficacy of feature selection methods, namely PSO and HS, on STFT data. The parameters from the Friedman test are X^2^r statistic, *p*-value, and its significance for *p* < 0.01, confidence level. 

## 3. Classification Using Machine Learning Techniques

This research employs various machine learning classifiers for lung cancer classification. The choice of the classifier is very significant in the classification methodology. Martín et al. [23] employed a Nonlinear Regression classifier for heterogeneous and highly nonlinear datasets related to the manufacturing process. For this research problem, such a classifier can capture complex nonlinear dependencies between gene expression levels and class labels is suitable. Linear classifiers cannot effectively capture these complex patterns and nonlinear decision boundaries. Also, as discussed in Dalatu et al. [24], Nonlinear regression models can be more robust to outliers than linear classifiers. Ficklin et al. [25] have utilized Gaussian Mixture Models (GMMs) for the classification of microarray gene data for various cancer types, namely lower-grade glioma, thyroid, glioblastoma, ovarian, and bladder cancer. Since GMMs are probabilistic models, they can work effectively on microarray gene expression datasets that are combinations of multi-modal distributions. The GMMs can capture complex data distributions by representing them as a combination of Gaussian components. Further, each Gaussian component in the mixture model can represent a different class, and the combination of these components enables the modeling of nonlinear decision boundaries that can be particularly useful when large amounts of samples overlap among the classes. Shah et al. [26] used the Softmax Discriminant Classifier for lung classification from microarray gene expression. The classifier works on estimating probability for each class, and it produces a probability distribution over all possible classes. This property is useful and suitable for multi-class classification problems, such as microarray gene expression data. The Softmax Discriminant Classifier is also computationally efficient as it incorporates regularization techniques to prevent overfitting and improve generalization. Ahmed et al. [27] used Naive Bayesian Classifier (NBC) to classify head and neck cancer from microarray gene expression data. NBCs are based on probabilistic principles, specifically Bayes’ theorem. The classifier computes the probabilities of diverse classes based on the feature values and allocates the class label with the highest probability. NBC framework also facilitates the integration of prior knowledge or expert beliefs through prior probabilities. Moreover, NBCs are computationally efficient, making them particularly suitable for large-volume and sparse datasets like microarray gene expression datasets. Vanitha et al. [28] used a Support Vector Machine (SVM) for gene expression-based data classification. SVM classifiers work on Vapnik’s statistical learning theory and thus handle the curse of dimensionality problem by finding the optimal hyperplanes in high-dimensional feature space. This process maximizes the margin between classes and provides robustness to outliers. In SVM, the mapping of higher dimensional feature space is achieved through the kernel trick. SVMs often yield sparse solutions and select the most informative genes contributing to the classification task. The sparse solution reduces the computational burden and improves generalization and classification accuracy. All of the above features of SVM make it a suitable classifier for microarray gene expression datasets. The following sections discuss the classification methodology of each classifier mentioned above.

### 3.1. Nonlinear Regression (NLR)

The nonlinear regression classification is another variant of the linear regression classifier that associates variables in the form of y = mx + c. Unlike linear regression models, nonlinear regression models associate variables by randomly changing the variable y. NLR strives to minimize the MSE of the individual observations in the dataset. The data points are adapted to a nonlinear function through multiple iterations for building a classifier model. The Euclidean distance is initially calculated from the dataset’s target and input data through the following expression as in Dai et al. [29].
(4)$$\sum y={\left\|T_{j}-X_{j}\right\|}^{2}$$

Here, *T* represents target data, and *X* represents input data with *j* as the sample index. The Euclidean distance is then projected to a three-dimensional space using the following cuboid equation:Minimize: z = k_1_ × y + k_2_^2^ × y^2^ + k_3_^3^ × y^3^(5)
(6)$$\mathrm{Subject}\;\mathrm{to}:\;\left\{\begin{array}{c} k_{1\;}>k_{2\;}>k_{3\;}>0,\;k_{i\;}[0,1]\;for\;i=1,2\;and\;3\\ {\left(\frac{k_{1\;}-k_{2}}{2}\right)}^{2}<0.5\\ k_{2\;}=\frac{k_{1\;}}{10},k_{3\;}=\frac{k_{1\;}}{10} \end{array}\right\}$$

Later, the minimum value of the three-dimensional space, f = min (z) is computed.

The threshold function ‘g’ for the NLR classification problem is framed with the help of the minimum value of the three-dimensional space, f = min (z) and d_0_ as the sum of the squares of mean deviation values as
g = f + d_0_(7)

Since the calculation of d_0_ involves least squares method, the methodology employed here may be also called as a least square based NLR.

### 3.2. Gaussian Mixture Model

The Gaussian Mixture Model (GMM) is a probabilistic model for classification and clustering tasks. The model considers that the data points are generated from a mixture of Gaussian distributions. The probability Density Function (PDF) of the Gaussian distribution having mean μ and covariance Ԑ is given by Ge et al. [30].
(8)$$p\left(x\right)=\frac{1}{\sqrt{{(2\pi )\;}^{n}\left|\varepsilon \right|}}\exp(-\frac{1}{2}{\left(x-\mu \right)}^{T}{Ԑ}^{-1}(x-\mu ))$$
where *x* is the data point, *n* hold the dimensionality of the data, and *T* indicates the transpose operation. The GMM assumes that the observed data are generated from a mixture of K Gaussian distributions. The posterior probability of the *k*th Gaussian component with mixing coefficient π*_k_* is given by:(9)$$p\left(k/x\right)=\frac{\pi_{k}\;\eta \;(x\;;\;\mu_{k\;\;\;},\;{Ԑ}_{k\;})\;}{\sum_{i-1}^{k}\pi_{i\;}\eta \;(x\;;\;\mu_{i\;\;\;},\;{Ԑ}_{i\;})}$$

µ*_k_* and Ԑ*_k_* are the covariance matrix and mean vector for the *k*th component with *k* as total number of components in the mixture. In the next step, the parameters of the GMM, including the mixing coefficients, means, and covariances, will be estimated using the Maximum Likelihood Estimation (MLE) technique. In the next step, the parameters of the GMM, including the mixing coefficients, means, and covariances, will be estimated using the Maximum Likelihood Estimation (MLE) technique. The MLE estimation for the GMM is defined as the log-likelihood of the observed data.
(10)$$Log\;L\;\left(\theta \right)=\sum_{i=1}^{N}log\left(\sum_{k=1}^{K}\pi_{k}\;\;\eta \;(x\;;\;\mu_{k},\;{Ԑ}_{k\;})\right)$$
where *θ* represents the set of all parameters of the GMM and *N* is the total number of observed data points. In the next step, the expectation minimization is performed through an iterative approach by computing posterior probabilities *p*(*k*/*x*) for each data point and updating parameters *θ* by maximizing the log-likelihood, considering the computed posterior probabilities.

### 3.3. Softmax Discriminant Classifier

The Softmax Discriminant Classifier uses the softmax function to transform the discriminant scores into class probabilities. At first, in SDC, the discriminant scores are calculated by taking the dot product between the feature vector *x* and the weight vector *w_k_* for each class *k*, along with the bias term *b_k_*. Mathematically, the discriminant score *z_k_* for class *k* is given by Hastie et al. [31]
(11)$${\;z}_{k}={w_{k}}^{T}x+b_{k}$$

For the discriminant scores *z* = [*z*_1_, *z*_2_… *z_k_*] for *K* classes, the softmax function calculates the probability of the input *x* belonging to class *k* using the following expression
(12)$$P\;(y=k/x)=\frac{\exp\left(z_{k}\right)}{\sum_{i=1}^{k}\exp\left(z_{i}\right)}$$

The loss function measures the discrepancy between the predicted class probabilities and the true class labels. The typically used loss function for Softmax Discriminant Classification is the cross-entropy loss. For the training dataset with *N* samples, the cross-entropy loss ‘*L*’ is defined as
(13)$$L=-\frac{1}{N}\sum_{i=1}^{N}\sum_{k=1}^{K}{y_{k}}^{\left(i\right)}log(P(y=k/x^{\left(i\right)}))$$
where *N* is the number of samples in the training dataset, *y_k_*^(*i*)^ is the true label for sample *i*, which is 1 if the sample belongs to class *k* and 0 otherwise, and $P(y=k/x^{\left(i\right)})$ is the predicted probability of class *k* for sample *i*. Finally, the weight vector $w_{k}$ and bias term $b_{k}$ are updated using gradient descent optimization method with learning rate α as follows.
(14)$$w_{k}=w_{k}-\alpha \;\frac{\partial \;L}{\partial w_{k}}$$
(15)$$b_{k}=b_{k}-\alpha \;\frac{\partial \;L}{\partial b_{k}}$$

### 3.4. Naive Bayesian Classifier (NBC)

The Naive Bayesian classifier (NBC) is efficient and straightforward and works on Bayes principles. NBC is a probabilistic classification algorithm based on the Bayes theorem and feature independence assumption. First, NBC calculates the posterior probability of a class using the prior probability and likelihood. For a given class *C* and features *x*_1_, *x*_2_… *x_n_*, posterior probability $p\left(C|x_{1}x_{2},\ldots \;,x_{n}\right)$ is expressed as Berrar [32].
(16)$$p\left(C|x_{1}x_{2},\ldots .\;,x_{n}\right)=\frac{p\left(C\right)\;.\;\;p\;(x_{1}x_{2},\ldots .\;,x_{n}\;|C)}{p\;(x_{1}x_{2},\ldots .\;,x_{n})}$$

For the class *C*, *p*(*C*) represents the prior probability, $p\;(x_{1}x_{2},\ldots .\;,x_{n}\;|C)$ is the likelihood, and $p\;(x_{1}x_{2},\ldots .\;,x_{n})$ is the evidence probability.

In Naive Bayes approach is that the features are conditionally independent for the class. This assumption that simplifies the calculation of the likelihood can be expressed as
(17)$$p\left(C|x_{1}x_{2},\ldots .\;,x_{n}\right)=p\left(x_{1}|C\right).p\left(x_{2}|C\right).\ldots \ldots \ldots .\;.\;p\left(x_{n}|C\right)$$
where $p\left(x_{i}|C\right)$ is the probability of feature $x_{i}$ of the class *C*. The $\;p\left(x_{i}|C\right)$ is estimated from the fraction of class *C* training examples with the feature value $x_{i}$. Then the prior probability $p\left(C\right)$ can be estimated as the fraction of training examples belonging to class *C*. Finally, to predict the class label for the features $x_{i}$, the algorithm calculates the posterior probability for each class and assigns the instance to the class with the highest probability.

### 3.5. Support Vector Machine Classifier (Linear)

In machine learning, Support Vector Machine (SVM) is a robust learning algorithm used for analyzing and classifying data. The SVM with a linear kernel model is the type of SVM that can solve linear problems by creating a line or hyperplane which separates the data into classes. The data that reside close to the hyperplane is considered the support vector. The expression for the hyperplane used in the binary classification problem can be written as Zhang et al. [33].
(18)$$f\left(x\right)=w^{T}x+b$$

Here, *f*(*x*) is the objective function with *x* as the feature vector, *w* as the weight vector perpendicular to the hyper plane and *b* as the bias term, determining the offset of the hyper plane from the origin. SVM aims to find the hyperplane that maximizes the margin. The optimization problem for SVM linear classification with margin $\frac{1}{\left\|w\right\|}$ is given by:(19)$$maximize\;\frac{1}{2}{\left\|w\right\|}^{2}\;\mathrm{Subject}\;\mathrm{to}\;{y_{i}(w}^{T}x_{i}+b)\geq 1\;\mathrm{for}\;\mathrm{all}\;\left(x_{i},y_{i}\right)\in \mathrm{training}\;\mathrm{data}$$

Here, $y_{i}$ represents the class label (+1 or −1) for the training example with feature vector $x_{i}$.

From the above primal optimization problem, the dual optimization problem is derived by finding the Lagrange multipliers or dual variables for each training example. The dual problem reduces computational burden using optimal Lagrange multipliers $\alpha_{i}$. Thus the SVM classification with the decision function *f_linear_*(*x*) is expressed as
(20)$$f_{linear}\left(x\right)=\sum_{i=1}^{N}\alpha_{i}y_{i}\left(x_{i\;}.\;x\right)+b$$

If *f_linear_*(*x*) is positive, the instance is assigned to one class, and if it is negative, it is assigned to the other class.

### 3.6. Support Vector Machine Classifier (Polynomial)

SVM with a polynomial kernel is a variant of SVM that allows for nonlinear classification by mapping the input features into a higher-dimensional space using a polynomial function. The polynomial kernel function calculates the dot product between the mapped feature vectors in the higher-dimensional space. The polynomial kernel function *K_poly_*(*x*,*z*) with *x* and *z* are pair of feature vectors is given by Zhang et al. [33].
(21)$$K_{poly}\left(x,z\right)={\left(\gamma \;x^{T}z+r\right)}^{d}$$
where *γ* is the coefficient scaling the dot product, *r* is the coefficient representing the independent term, and *d* is the degree of the polynomial. Further, the decision function in SVM polynomial classification is defined as the linear combination of the kernel evaluations between the support vectors and the input feature vector with the bias term. The decision function $f_{poly}\left(x\right)$ is given by:(22)$$f_{poly}\left(x\right)=\sum_{i=1}^{N}\left(\alpha_{i}y_{i}\right)K_{poly}\left(x_{i},x\right)+b$$

Based on *f_poly_*(*x*) value, the class assignment is performed in a similar way as performed in SVM (Linear) classifier.

### 3.7. Support Vector Machine Classifier (RBF)

The Radial Basis Function (RBF) can also be used as a kernel while using SVM classifier. Here, nonlinear mapping of the input features into a higher-dimensional space is performed using an RBF. The RBF kernel *K_RBF_*(*x*,*z*) that is used to compute the similarity between feature vectors in the input space is expressed as Zhang et al. [33].
(23)$$K_{RBF}\left(x,z\right)=\exp\left(-\frac{{\left\|x-z\right\|}^{2}}{2\sigma^{2}}\right)$$
where |*x* − *z*| is the Euclidean distance between *x* and *z* with *σ* as the kernel width parameter that controls the influence of each training sample. As computed before, the decision function for SVM RBF classification is also defined as a linear combination of the kernel evaluations between the support vectors and the input feature vector with the bias term. The decision function *f_RBF_*(*x*) for SVM RBF is given by:(24)$$f_{RBF}\left(x\right)=\sum_{i=1}^{N}\left(\alpha_{i}y_{i}\right)K_{RBF}\left(x_{i},x\right)+b$$

Based on *f_RBF_*(*x*) value, the class assignment is performed in a similar way as performed in SVM linear and polynomial classifiers.

### 3.8. Selection of Target

This research involves a binary classification, and hence two targets *T_Adeno_* and *T_Meso_* for Adeno and Meso classification and mapping constraints must be selected. The target and mapping constraints are selected based on the distribution of classes in the dataset and the class imbalance problems involved in the microarray gene expression dataset. The target value, $T_{Adeno}\;\in [0,1]$ is having the following constraint. For *N* number of features, with *µ_i_* as the average of input feature vector, the mapping constraint is framed as:(25)$$\frac{1}{N}\sum_{i=1}^{N}\mu_{i}\leq T_{Adeno}$$

*T_Adeno_* must have a target value greater than *µ_i_* and the average of input feature vectors *µ_j_* for the Meso case. The target value $T_{Meso}$ ∈ [0, 1], for the Meso case with *M* number of features, the mapping constraint is framed as:(26)$$\frac{1}{M}\sum_{j=1}^{M}\mu_{j}\leq T_{Meso}$$

Also, the difference between the target values must follow the following constraint:(27)$$||T_{Adeno}-T_{Meso}||\geq 0.5$$

Thus, following all of the above mentioned constraints, the targets *T_Adeno_*, and *T_Meso_* for Adeno and Meso output classes are chosen as 0.85, and 0.65, respectively. The performance of classifiers is will be evaluated based on the Mean Squared Error (MSE).

### 3.9. Training and Testing of Classifiers

The training data for the dataset are limited. So, we perform k-fold cross-validation. K-fold cross-validation is a popular method for estimating the performance of a machine learning model. The process performed by Fushiki et al. [34] for k-fold cross-validation is as follows. The first step is to divide the dataset into k equally sized subsets (or “folds”). For each fold, i, train the model on all of the data except the i-th fold and test the model on the i-th fold. The process is repeated for all k folds so that each is used once for testing. At the end of the process, you will have k performance estimates (one for each fold). Now, calculate the average of the k performance estimates to obtain an overall estimate of the model’s performance. Once the model has been trained and validated using k-fold cross-validation, you can retrain it on the full dataset and predict new, unseen data. The advantage of k-fold cross-validation is that it provides a more reliable estimate of a model’s performance than a simple train-test split, as it uses all of the available data. In this paper, the k-value is chosen as 10-fold. This research used a value of 2049 dimensionally reduced features per patient. This research is associated with 150 patients of Adeno Carcinoma and 31 Meso cancer patients with multi-trail training of classifiers required. The use of cross-validation removes any dependence on the choice of pattern for the test set. The training process is controlled by monitoring the Mean Square Error (MSE), which is deduced from Fushiki et al. [34] as,
(28)$$MSE=\frac{1}{N}\sum_{j=1}^{N}{\left(O_{j}-T_{j}\right)}^{2}$$
where *O_j_* is the observed value at time *j*, *T_j_* is the target value at model *j*; *j* = 1 and 2, and *N* is the total number of observations per epoch. In our case, it is 2049.

Table 3 shows the confusion matrix for lung cancer detection.

In the case of lung cancer detection, the following terms can be defined:True Positive (TP): A patient correctly identifies as having Adeno Carcinoma lung cancer.True Negative (TN): A patient is correctly identified as having Meso cancer.False Positive (FP): A patient is incorrectly identified as having Adeno Carcinoma lung cancer when they have Meso Carcinoma cancer disease.False Negative (FN): A patient is incorrectly identified as having Meso Carcinoma lung cancer when they do have the Adeno Carcinoma cancer disease.

Table 4 explores the training and testing MSE performance of the classifiers without and with feature selection (PSO and Harmonic Search) Methods for STFT Dimensionality reduction techniques. The standard training MSE always varies between 1 × 10^−6^ and 1 × 10^−8^, while the standard testing MSE varies from 1 × 10^−4^ to 1 × 10^−7^. GMM classifier without feature selection method settled at minimum training and Testing MSE of 3.6 × 10^−8^ and 7.21 × 10^−7^, respectively. PSO Feature Selection method SVM (RBF) Classifier scores minimum training and Testing MSE of 2.25 × 10^−9^ and 3.6 × 10^−8^, correspondingly. Like the Harmonic Search feature Selection method, the SVM (RBF) classifier once again attained minimum Training and Testing MSE of 2.56 × 10^−8^ and 1.96 × 10^−7^, respectively. The minimum testing MSE is one of the indicators towards the attainment of better classifier performance. As shown in Table 4, the higher the value of testing MSE leads to poorer performance of the classifier, irrespective of the feature selection methods.

Table 5 explores the selection of classifier parameters during the training process. The classifiers parameters are chosen on a trial and error basis for minimum training MSE training condition. The stopping criteria for training MSE is fixed to either minimum standard MSE value or 1000 iterations. It is observed that the training MSE value reached 1.0 × 10^−10^ within 1000 iterations as the training progressed.

## 4. Classification Using CNN Methods

Convolutional Neural Networks (CNNs) are a deep learning model commonly used for image classification tasks. The CNN classification methodology adopted in this research is depicted in Figure 7. The CNN architecture employed in this research consists of a convolutional layer, a fully connected layer and a softmax layer. The convolutional layer is the building block of CNN that performs a convolution operation on the input data using a set of learnable filters. The convolutional layer consists of a convolutional 2D layer, Batch Normalization Layer, ReLU layer, and Max Pooling 2D layer. The classification methodology in CNNs involves stacking these layers in a sequential manner. Combining these layers and training the network, CNNs can effectively learn and classify complex patterns in image data.

In the convolutional 2D layer, the convolution operation is performed between the filter and local patches of the 12,533 × 181 input data. This stage produces feature maps and patterns that help to improve the stability and convergence of the network during training. Next, the batch normalization layer normalizes the mean and variance of the input by utilizing mini-batches of training data from the Max Pooling 2D layer. The Batch Normalization Layer helps in reducing internal covariate shift and allows the network to learn more efficiently. The ReLU Layer applies the Rectified Linear Unit (ReLU) activation function element-wise to the Batch Normalization layer’s output. The ReLU function sets negative values to zero and keeps positive values unchanged. This step is a crucial process that introduces nonlinearity into the network, enabling CNN to learn complex and nonlinear relationships in the dataset. The final step in the convolutional layer is a 2 × 2 Max Pooling 2D layer, which reduces the feature maps’ spatial dimensions while retaining the most prominent features. Max Pooling 2D layer divides the input into non-overlapping regions and only keeps the maximum value from each region. Thus, Max pooling helps in reducing computational complexity and controlling the over fitting problem. Likewise, in this research, we employed three convolutional layers with filter sizes 16, 32 and 64.

The convolutional layers learn local image features, while the pooling layers reduce spatial dimensions. Batch normalization and ReLU layers help in normalization and nonlinearity, respectively. The fully connected layers capture high-level representations, and the softmax layer produces class probabilities for classification. By combining these layers and training the network using techniques like back propagation and optimization algorithms, CNNs can effectively learn and classify complex patterns in image data.

### 4.1. CNN Training and Testing Parameters

The optimal training and testing parameter values are set using trial and error methods to achieve better classification accuracy and improved CNN performance. The ten-fold cross-validation technique averages the performance across multiple validation sets. The dataset is divided into ten folds; each used as a validation set once. The remaining nine folds are used for training. The process is repeated ten times with a different fold as the validation set. As the dataset is imbalanced, the cross-validation techniques prevent overfitting and underfitting problems by representing the distribution of the minority and majority classes over every fold. The learning rate is selected to prevent overshooting and obtain an optimal set of weights for the CNN model. The mini-batches that use simultaneous processing of a subset of training samples provide efficient computation, and better generalization, reducing memory requirements and allowing for parallelization. The maximum number of epochs is fine-tuned to ensure the model has sufficient opportunities to learn the complex microarray gene dataset patterns while avoiding the risk of overfitting and under fitting. The number of convolutional and fully connected layers is chosen such that the CNN model captures intricate patterns and achieves higher accuracy in classification. The filter size is tuned to obtain the spatial relationships and extract different levels of features. The number of filters is selected to learn relevant patterns and representations in the microarray gene expression data. From all of the above observations, the optimal training and testing parameter values are fixed and are given in Table 6.

The cross-entropy loss is a popular loss function used with CNNs for training the model. The cross-entropy loss measures the difference between the predicted class probabilities and the true class labels. It enables the CNN to minimize the difference between the predicted and actual distributions. The cross entropy loss ‘L’ in CNN is defined as:(29)$$L\left(y,p\right)=-\sum_{i}y_{i}log(p_{i})$$
where y is the one-hot encoded true class label vector, and p is the predicted class probability vector. The cross-entropy loss is significant for CNNs as it predicts how well the class probabilities match the ground truth labels. Also, the network learns meaningful representations and makes accurate predictions by minimizing the cross-entropy loss. Also, in CNN, the gradients obtained from the cross-entropy loss facilitate efficient parameter updates during back propagation.

### 4.2. Accuracy

The accuracy of a classifier is a measure of how well it correctly identifies the class labels of a dataset. It is calculated by dividing the number of correctly classified instances by the total number of instances in the dataset. The equation for accuracy is given by Wilson et al. [35].
(30)$$Acc=\frac{\left(TN+TP\right)}{\left(TN+FN+TP+FP\right)}$$

A sample of the accuracy and loss function during CNN training and testing progress is shown in Figure 8. The accuracy function measures the percentage of correctly classified samples to determine the CNN model’s performance. It also helps to monitor training progress and evaluate the model’s ability to generalize unseen data. On the other hand, the loss function quantifies the discrepancy between the predicted and actual outputs. By optimizing the loss function, the CNN model can adjust its parameters to minimize errors and learn the underlying patterns in the data.

Table 7 exhibits the training and testing performance of the classifiers based on the accuracy of the raw data and with STFT features for CNN Methods. As in the case of raw data, the training accuracy of classifiers always varied between 87.235% and 93.631%, while the testing accuracy varied from 83.425% to 90.607%. In the STFT feature method, Naïve Bayesian Classifier arrived at maximum training accuracy of 94.452%. The testing accuracy is ebbed at the peak value of 91.66% for the classifiers SDC and Naïve Bayesian. Table 7 shows that the nonlinear regression classifier is a poor-performing one for both cases of the inputs.

To analyze the statistical significance of training and testing process for CNN with Raw Data and STFT features, the mean and standard deviation of error is calculated. The mean of the standard error for training accuracy of CNN methods with Raw Data is obtained as 1.758%, and mean standard deviation of error attained as 0.324%. Likewise, the mean of the standard error for testing accuracy of CNN methods with Raw Data is obtained as 2.094%, and mean standard deviation of error attained as 0.568%. The mean of the standard error for training accuracy of CNN methods with STFT features is obtained as 1.565%, and mean standard deviation of error attained as 0.3721%. Likewise, the mean of the standard error for testing accuracy of CNN methods with STFT features is obtained as 1.952%, and mean standard deviation of error attained as 0.4688%. Overall, it is observed from the Table 7 that, the training and testing error is less than 2%, and standard deviation is less than 1%. Therefore, this implies that the error is insignificant when compared with the training and testing accuracy values.

As we evaluate Table 7 with minimum and maximum accuracy obtained for training and testing for the two CNN methods (Raw Data and STFT Features) by adding and subtracting the standard error values, the following inferences are arrived at. For the CNN method with Raw Data input, the SVM (RBF) attained the better training accuracy of 92.221% and 95.041% with the mean accuracy ± standard error. In the testing case, the SVM (RBF) attained a testing accuracy of 89.28773% for mean accuracy—standard error. At the same time, GMM attained the best testing accuracy of 92.24276% in mean accuracy + standard error condition. Likewise, for the CNN method with STFT Features, the Naïve Bayesian attained the better training accuracy of 93.0727% and 95.8327% with the mean accuracy ± standard error. In the testing case, the SDC attained the testing accuracy of 89.64667% for mean accuracy—standard error. At the same time, once again, Naïve Bayesian maintained the best testing accuracy of 94.11667% in mean accuracy + standard error condition. It is also observed that there is no change in the nonlinear regression classifier’s performance level.

## 5. Results and Discussion

The research utilizes formal ten-fold testing for machine learning and CNN classification methodologies. The training uses 85% of the gene expression data, and the remaining 15% are employed for testing the model. Since this research is binary classification, the performance metrics are accordingly chosen. In binary classification problems, a confusion matrix is a valuable tool for evaluating the performance of a machine learning model. As mentioned earlier, the confusion matrix summarizes the predictions made by the model against the true labels of the data. The performance metrics such as Accuracy, Precision, Recall, F1 score, MCC, Error Rate, and Kappa are derived by analyzing the values within the confusion matrix. Next, we discuss in detail the performance matrices employed in this research.



*Precision*




(31)
$$Precision=\frac{TP}{\left(TP+FP\right)}$$




*Recall*




(32)
$$Recall=\frac{TP}{\left(TP+FN\right)}$$




*F1 Score*



The F1 score is a measure of a classifier’s accuracy that combines precision and recall into a single metric. It is calculated as the harmonic mean of precision and recall, with values ranging from 0 to 1, where 1 indicates perfect precision and recall. The equation for F1 score is given by Koizumi et al. [36].
(33)$$F1=\frac{2\times TP}{\left(2\times TP+FP+FN\right)}$$

Here, precision is the proportion of true positives among all instances classified as positive, and recall is the proportion of true positives among all positive instances. The F1 score is useful when the classes in the dataset are imbalanced, meaning there are more instances of one class than the other. In such cases, accuracy may be a bad metric, as a classifier that predicts the majority class would have high accuracy but low precision and recall. The F1 score provides a more balanced measure of a classifier’s performance.
*Matthews Correlation Coefficient (MCC)*

MCC stands for “Matthews Correlation Coefficient”, which measures the quality of binary (two-class) classification models. It considers true and false positives and negatives and is particularly useful in situations where the classes are imbalanced.

The MCC is defined by the following equation as given in Chicco et al. [37]:(34)$$MCC=\frac{\left(TP\;*\;TN-FP\;*\;FN\right)}{\sqrt{TP+FP)\;*\;(TP+FN)\;*\;(TN+FP)\;*\;(TN+FN)}}$$

The MCC takes on values between −1 and 1, where a coefficient of 1 represents a perfect prediction, 0 represents a random prediction, and −1 represents a perfectly incorrect prediction.



*Error Rate*



As mentioned by Duda et al. [38], the error rate of a classifier is the proportion of misclassified instances. It can be calculated using the following equation:(35)$$Error\;rate=\frac{\left(FP+FN\right)}{\left(TP+TN+FP+FN\right)}$$
*Kappa*

The kappa statistic, also known as Cohen’s kappa, measures agreement between two raters or between a rater and a classifier. In the context of classification, it is used to evaluate the performance of a classifier on a binary or multi-class classification task. The kappa statistic measures the agreement between the predicted and true classes, considering the possibility of the agreement by chance. Cohen et al. [39] defined kappa as follows:(36)$$Kappa=\frac{{(P}_{o}-P_{e})}{\left(1-P_{e}\right)}$$
where $P_{o}$ is the observed proportion of agreement, and $P_{e}$ is the proportion of agreement expected by chance. $P_{o}$ and $P_{e}$ are calculated as follows:(37)$$P_{o}=\frac{\left(TP+TN\right)}{\left(TP+TN+FP+FN\right)}$$
(38)$$P_{e}=\frac{\left(TP+FP)\;*\;(TP+FN)+(FP+TN)\;*\;(FN+TN\right)}{{\left(TP+TN+FP+FN\right)}^{2}}$$

The Kappa is always less than or equal to 1. The Kappa coefficient with a value of 1 implies perfect agreement and any value less than 1 can be interpreted as follows: Poor agreement (<0.2), Fair agreement (0.2–0.4), Moderate agreement (0.4–0.6), Good agreement (0.6–0.8), and Very good agreement (0.8–1). The results are tabulated in the following tables.

Table 8 indicates the performance analysis of the classifiers based on parameters such as Accuracy, Precision, Recall, F1 Score, MCC, Error Rate, and Kappa values for the STFT Dimensionality Reduction method without feature selection methods. It is explored from Table 8 that the GMM Classifier is a high performing one with an accuracy of 80.66%, Precision of 92.59%, Recall of 83.33%, an F1 Score of 87.71%, with a low error rate of 19.34%. The GMM Classifier also demonstrates a moderate value of MCC 0.4419 and a Kappa value of 0.4285. The Softmax Discriminant Classifier is a low-performing classifier with a low accuracy of 59.11%, with high Error Rate of 40.89%, Precision of 90.42%, Recall of 56.66%, and an F1 Score of 69.67%. The MCC and Kappa values of the SD classifier are 0.2083 and 0.1609, respectively.

Table 9 depicts the performance analysis of the classifiers for the STFT Dimensionality Reduction method with PSO feature selection methods. It is identified from Table 9 that the SVM (RBF) Classifier achieved high accuracy of 94.47%, Precision of 97.94%, Recall of 95.33%, and an F1 Score of 96.62% with a low error rate of 5.52%. The SVM (RBF) Classifier has also reached a high value of MCC 0.81709 and Kappa value of 0.81485. The nonlinear regression classifier is placed at the lower edge with a low accuracy of 59.91%, a high Error Rate of 43.09%, Precision of 88.297%, Recall of 55.34%, and an F1 Score of 68.03%. The MCC and Kappa values of the nonlinear Regression classifier are at 0.1496 and 0.1156, correspondingly.

Table 10 exhibits the performance analysis of the classifiers for the STFT Dimensionality Reduction method with Harmonic Search feature selection methods. As shown in Table 10, the SVM (RBF) Classifier achieved high accuracy of 90.05%, Precision of 97.82%, Recall of 90%, F1 Score of 93.75% with a low error rate of 9.94%. The SVM (RBF) Classifier is also maintained at a high value of MCC 0.711 and Kappa value of 0.6963. Unfortunately, SVM (poly) classifier is placed at the low performing one with an accuracy of 59.11%, Precision of 88%, Recall of 58.67%, and a high Error Rate of 40.89% and an F1 Score of 70.4%. The MCC and Kappa value of the SVM (Poly) classifier is at 0.1512 and 0.1217, accordingly.

Figure 9 shows the Performance of Classifiers with and without Feature Selection regarding MCC and Kappa parameters. As indicated by Figure 9 that the SVM (RBF) Classifier is reached a high value of MCC 0.81709 and Kappa value of 0.81485. The nonlinear Regression Classifier for the PSO feature selection method attains low MCC and Kappa values of 0.1496 and 0.1156. The average MCC and Kappa values across the classifiers for feature selection are 0.3212 and 0.27403, respectively. The average MCC and Kappa values across the classifiers for PSO and Harmonic Search Feature selection methods are settled at 0.4194, 0.3878 and 0.4841 and 0.4667. This signature effect of Feature selection shows the improvement of average MCC and Kappa values across the classifiers.

Figure 10 shows the Performance of Classifiers with and without Feature Selection regarding parameters like Accuracy, F1 Score and Error Rate. As identified by Figure 10, the SVM (RBF) Classifier with the PSO feature selection method achieved higher values in Accuracy of 94.47%, an F1 Score of 96.62% and a lower Error Rate of 5.52% than all other categories of classifiers. In the case of the Harmonic Search feature selection method, once again SVM (RBF) classifier attained good values of Accuracy of 90.05%, an F1 Score of 93.75% and a low Error Rate of 9.944% compared with all other six classifiers. GMM Classifier attained an appreciable value of Accuracy of 80.66%, F1 Score of 87.71% and moderate Error Rate of 19.33% for the STFT inputs without any feature selection methods. The effect of feature selection improves the classifiers’ benchmark parameters and overall performance. The PSO feature selection retained its top position. It maintained its superiority over the harmonic search feature selection method, reflected in the classical improvements of the classifier’s performance.

Table 11 explores the performance analysis of the classifiers for raw microarray gene data with CNN methods. As displayed in Table 11, the SVM (RBF) Classifier achieved the highest accuracy of 90.607%, Precision of 90.797%, Recall of 98.67%, and an F1 Score of 94.56%, with a low Error Rate of 9.39%. The SVM (RBF) Classifier is also maintained at a moderate value of MCC 0.6329 and Kappa value of 0.6031. For the CNN method, all seven classifiers are maintained at more than 83% accuracy, Precision of 88.554%, Recall of 98%, and more than 90% F1 Score. The Softmax Discriminant Classifier attained minimum MCC and Kappa values of 0.5016 and 0.4616, respectively.

Table 12 exhibits the performance analysis of the classifiers for the STFT Dimensionality Reduction method with CNN methods. Table 12 shows that Softmax Discriminant Classifier (SDC) achieved a high accuracy of 91.66%, Precision of 93.54%, Recall of 96.67%, an F1 Score of 95.08%, with a low error rate of 8.33%. The SD Classifier is also maintained at a high MCC value of 0.6825 and Kappa value of 0.6785. All seven classifiers are maintained at more than 86% of accuracy, Precision of 87%, Recall of 96%, and more than 92% of F1 Score. The Naïve Bayesian Classifier achieved an accuracy of 91.66%, Precision of 90.909%, Recall of 100%, and also attained MCC and Kappa values of 0.6742 and 0.625, respectively. It is also observed from Table 12 that STFT input to the CNN method enhances the performance metrics of the classifiers when compared with raw input with CNN methods, as discussed in the paper.

Figure 11 depicts the Performance of Classifiers regarding MCC and Kappa parameters for Raw and STFT inputs to the CNN method. As indicated by Figure 11, the SD Classifier for the STFT feature method reached high MCC and Kappa values of 0.6825 and 0.6785, respectively. The average MCC and Kappa values across the classifiers are 0.5236 and 0.485. As shown by Figure 11 that the SVM (RBF) Classifier is attained at good values of MCC 0.6329 and Kappa value of 0.6031 for the raw inputs to the CNN method. The average MCC and Kappa value across the classifiers for CNN is 0.4794 and 0.4226, respectively. The Scalogram effect of STFT inputs to the CNN methods exemplifies the enhancement of average MCC and Kappa values across the classifiers.

Figure 12 shows the Performance of Classifiers regarding Accuracy, F1 Score, and Error Rate parameters for raw and STFT inputs with the CNN method. As depicted by Figure 12, the SVM(RBF) Classifier for raw input with the CNN method reached high accuracy, F1 Score and low Error Rates of 90.6%,94.56% and 9.39%, respectively. As displayed in Figure 12 that the SD Classifier is attained at the high accuracy value, F1 Score and low Error Rate of 91.66%, 95.08% and 8.33%, respectively, for STFT inputs with the CNN method. The yardstick effect of STFT inputs in CNN methods maketh the improvement in the accuracy, F1 Score and Error Rate values across the classifiers.

Figure 13 displays the Performance of Classifiers in terms of Deviation of MCC and Kappa Parameters with mean values. The MCC and Kappa are the benchmark parameters that indicate the classifiers’ outcomes for different inputs. As in this research, there are two categories of inputs like raw microarray genes, STFT, STFT with PSO and Harmonic Search Feature selection, STFT with Scalogram method and finally, CNN methods are provided to the classifiers. The classifier’s performance is observed through the attained MCC and Kappa values for these inputs. The average MCC and Kappa values from the classifiers are 0.4681 and 0.44518, respectively. Now a methodology is devised to identify the deviation of MCC and Kappa values from their respective mean values to point out the classifier’s performance. It is also observed from Figure 13 that the MCC and Kappa values placed in the third quadrant of the graph depict the nonlinear outcome of the classifiers with lower performance metrics. The MCC and Kappa values placed at the first quadrant show higher outcomes for classifiers with MCC and Kappa values more than the average value. This specifies the classifier’s performance is improved for the STFT inputs along with the PSO Feature selection method. Figure 13 is also denoted by the curve fitting of the linear line with the following equation y = 1.062x + 1 × 10^−7^ and R^2^ = 0.991 values.

### Computational Complexity

The computational complexity analyzes the efficiency of the machine learning methods in terms of time and space complexity. In this research, the Big O notation represents the computational complexity of the dimensionality reduction, feature selection and classification methodologies. The computational complexity is represented by O (n), where n is the number of features. The O(log2n) means that the computational complexity increases ‘log2n’ times for any increase in ‘n’. The classifiers considered in this research are integrated with either dimensionality reduction techniques, feature selection techniques, or a combination of both. Therefore, the computational complexity also is a combination of these hybrid methods. Overall, the choice of methodology should consider the trade-off between computational complexity and the desired level of performance in classification methodology. Table 13 shows the computational complexity of the classifiers for the STFT dimensionality reduction method with and without feature selection methods, along with CNN Models.

As noted from Table 13, the SVM (Linear), NBC, Nonlinear Regression, Softmax Discriminant Classifiers, and SVM (RBF) classifiers are at the level of low computational complexity for the STFT DR method without feature selection methods. GMM Classifier attains moderate complexity of O (2n^4^ log2n) without feature selection methods and achieved good accuracy of 80.66%. For PSO and Harmonic search feature selection methods, the SVM (RBF) classifier with the computational complexity of O (2n^5^ log4n) placed at high accuracy of 94.47% and 90.05%, respectively. GMM and SVM (polynomial) classifiers are denoted by the high computational complexity of O (2n^7^ log2n) for the PSO and Harmonic Search Feature selection methods poorly performed in terms of accuracy score. These poor performances of classifiers are due to the presence of outlier problems in the STFT features. In order to enhance the performance of the classifiers, the STFT Scalogram features and CNN methods are included in this research. The SVM (RBF) Classifier attained good performance with moderate complexity of O (2n^3^ log4n) and O (2n^3^ log8n) for the methods, respectively. Next, through Table 14, we compare the accuracy of our research work with various methodologies across diverse microarray gene datasets employed for Adenocarcinoma and Mesothelioma lung cancer.

Table 15 compares previous works involving Lung and other types of cancer classification from microarray gene datasets with the reference from Fathi et al. [43] and Karmokar et al. [48], which contains performance metrics, namely Accuracy, Precision, Recall and F1 Score. Fathi et al. [43] utilized the PCC–DTCV (Pearson correlation–Decision Tree Cross Validation) model for lung cancer classification and reported accuracy in ranges from 93 to 95. The model involves a tuning parameter to maximize the depth of the decision tree classifier using grid search and Pearson correlation for the feature selection method. The grid search model is employed for dimensionality reduction and selects the most informative genes which involve higher computational complexity. In our research, we have utilized the gene expression intensity values incorporated with dimensionality reduction and classification. Unlike Fathi et al. [43], our methodology does not select prominent genes that increase computational overhead. Therefore, our method involves less computational complexity but is at an accuracy trade-off. Comparison with other types of cancer, namely prostate, breast, and leukemia, is also recorded.

Overall, this research evaluates and explores machine learning and CNN classifiers for detecting lung cancer from microarray gene expressions. The research investigation shows that although CNNs are good at learning features, due to their black-box nature, the CNNs fail to analyze the physical meaning behind forecasting and categorizing gene expression patterns from the learned features. Further, the LH2 dataset used for this research contains data imbalance, and many times during the analysis, CNNs were prone to overfitting and underfitting issues. The measures like the feature extraction step and replacement of the softmax layer with suitable classifiers were adopted to rectify these issues. However, the trade-off is that computational complexity will be increased while adding multiple stages to the CNN classification methodology.

Our research shows that microarray gene expression-based data are instrumental for disease classification. It contributes significantly to understanding gene expression patterns and their association with various biological processes and diseases. However, microarray genes have some limitations over general approaches like mRNA-Seq. The microarray gene expressions are selected from limited areas with specific genes or tissue regions. Therefore, the data generated may not capture the entire transcriptome and provide comprehensive information about gene expression. Also, microarray experiments may be prone to background noise and cross-hybridization problems due to non-specific binding, resulting in decreased sensitivity and specificity. Microarray experiments are expensive and sometimes have limited throughput compared to newer high-throughput sequencing technologies like mRNA-Seq.

## 6. Conclusions

The generation of malignant lung tissue is often inconspicuous and does not show symptoms in the early stages. Therefore the early detection of lung cancer is significant as it can lead to improved treatment outcomes and increased chances of survival. The microarray gene expression analysis of lung cancer data effectively detects lung cancer at an early stage, allowing for prompt intervention and potentially curative treatment options. Therefore, this paper uses microarray gene expression combined with machine and deep learning techniques to improve lung cancer data classification. The research employs two levels of pre-processing lung cancer data for better classification results: dimensionality reduction and feature selection techniques. The Short-Term Fourier Transform (STFT) dimensionality reduction with PSO feature selection and SVM (RBF) classifier produced the highest accuracy of 94.47%. The future endeavors of this research focus on improving the classification accuracy of the CNN classifiers by investigating nonlinear transforms like wavelets and nature-inspired feature extraction and selection methods incorporated into prior stages of CNN classification.

## Figures and Tables

**Figure 1 bioengineering-10-00933-f001:**
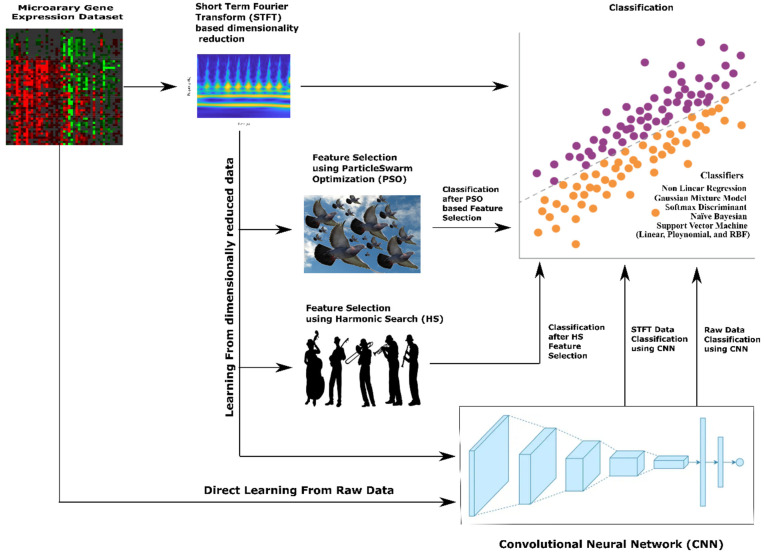
Methodology for Dimensionality Reduction and Classification of microarray gene expression data for lung cancer.

**Figure 2 bioengineering-10-00933-f002:**
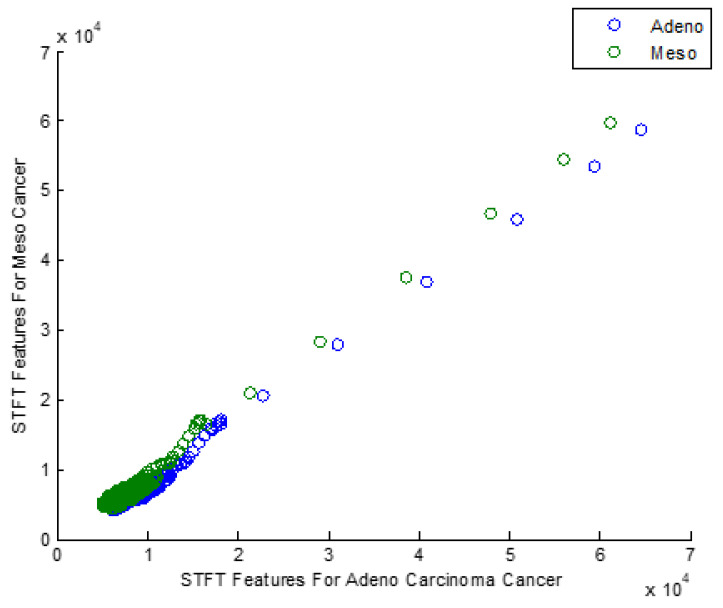
Scatter plot for STFT based Dimensionality Reduction Method in Meso and Adeno Carcinoma Cancer Classes.

**Figure 3 bioengineering-10-00933-f003:**
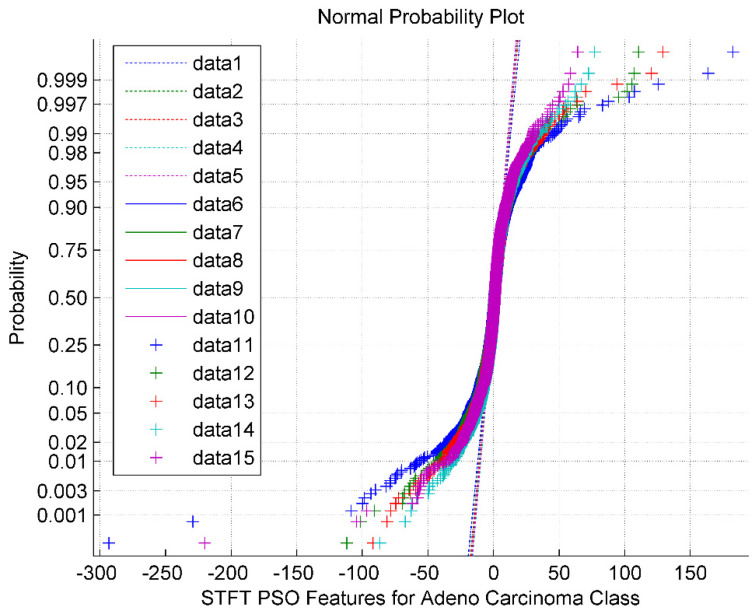
Normal Probability plot for STFT Dimensionality Reduction Method with PSO Feature Selection in Adeno Carcinoma Cancer Classes.

**Figure 4 bioengineering-10-00933-f004:**
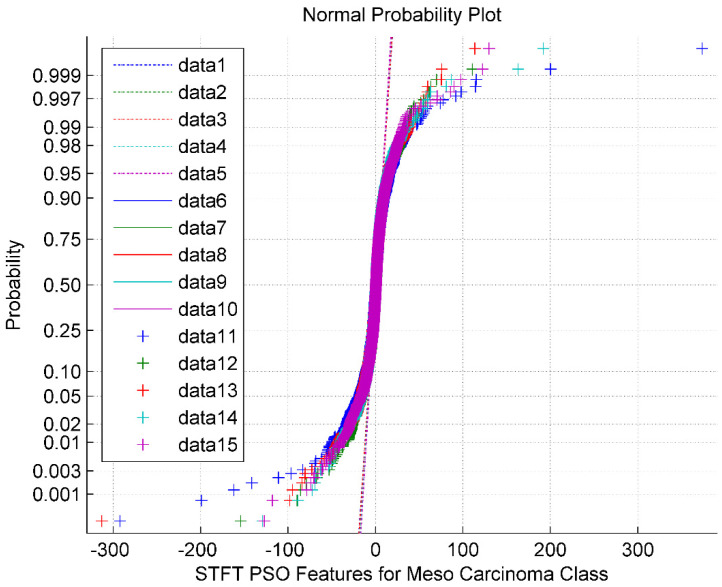
Normal probability plot for STFT Dimensionality Reduction Method with PSO Feature Selection in Meso Carcinoma Cancer Classes.

**Figure 5 bioengineering-10-00933-f005:**
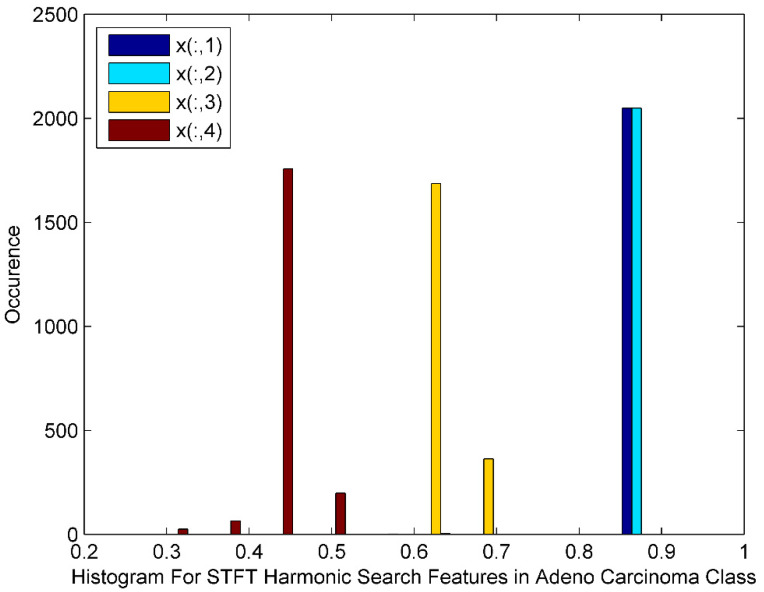
Histogram for STFT Dimensionality Reduction Method with Harmonic Search Feature Selection in Adeno Carcinoma Cancer Classes.

**Figure 6 bioengineering-10-00933-f006:**
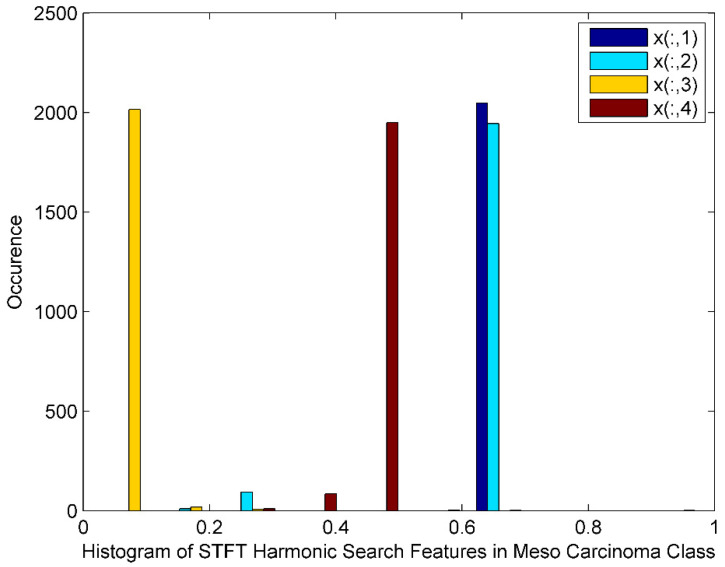
Histogram for STFT Dimensionality Reduction Method with Harmonic Search Feature Selection in Meso Carcinoma Cancer Classes.

**Figure 7 bioengineering-10-00933-f007:**
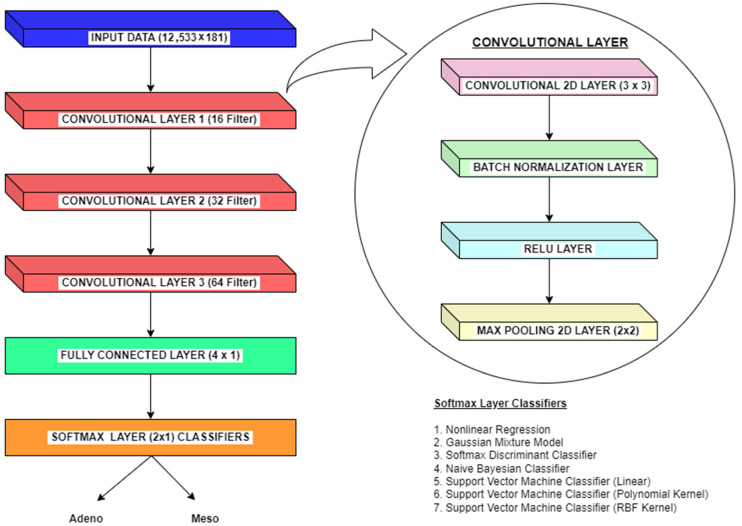
CNN Classification Methodology.

**Figure 8 bioengineering-10-00933-f008:**
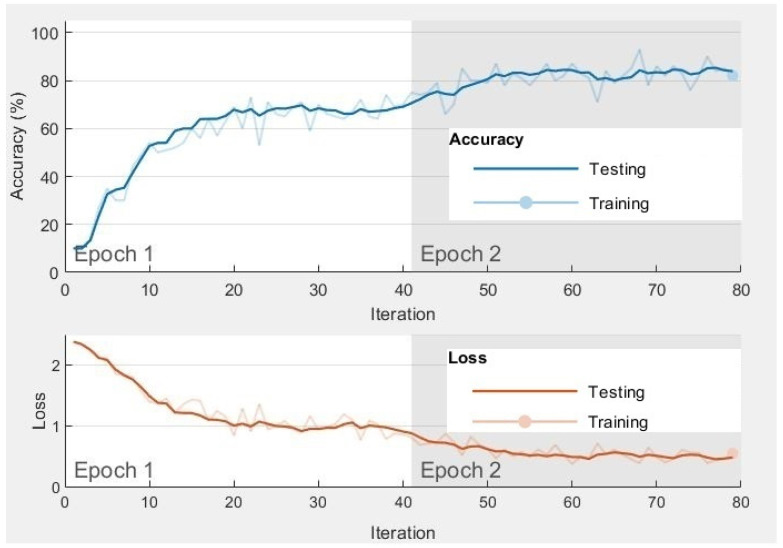
Training and testing progress of the first two epochs of SVM (RBF) Classifier in CNN Method with Raw Data.

**Figure 9 bioengineering-10-00933-f009:**
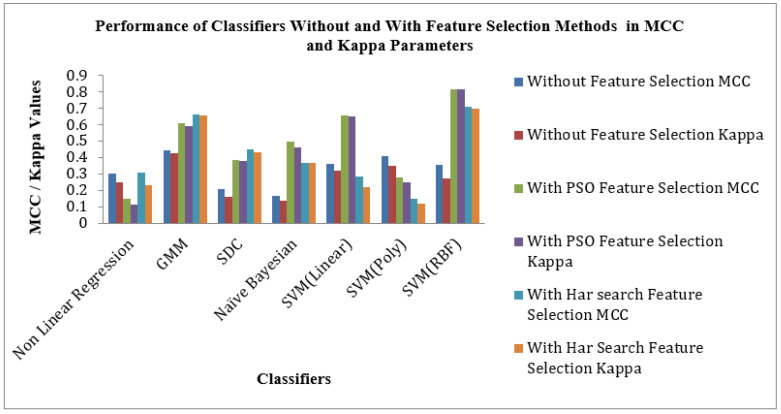
Performance of Classifiers Without and With Feature Selection Methods in terms of MCC and Kappa Parameters.

**Figure 10 bioengineering-10-00933-f010:**
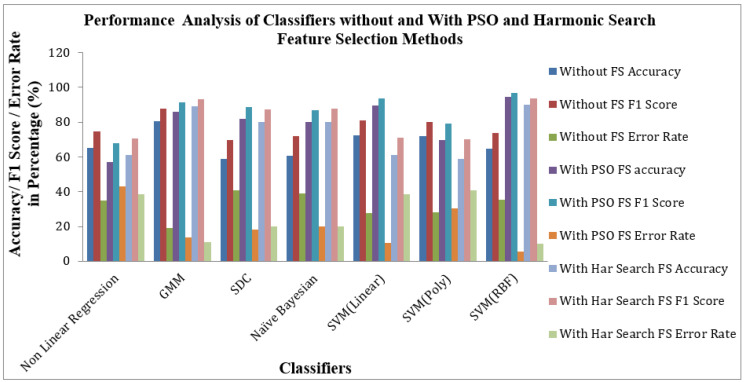
Performance of Classifiers Without and With Feature Selection Methods in terms of Accuracy, F1 Score and Error Rate Parameters.

**Figure 11 bioengineering-10-00933-f011:**
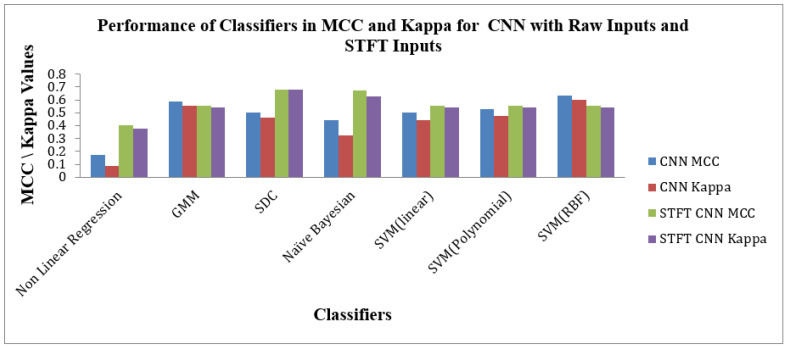
Performance of Classifiers in terms of MCC and Kappa Parameters for Raw and STFT Inputs for CNN Methods.

**Figure 12 bioengineering-10-00933-f012:**
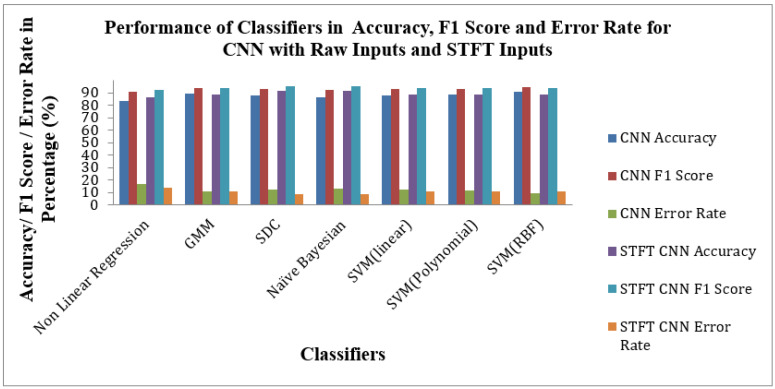
Performance of Classifiers in terms of Accuracy, F1 Score and Error Rate Parameters for Raw and STFT Inputs in CNN Methods.

**Figure 13 bioengineering-10-00933-f013:**
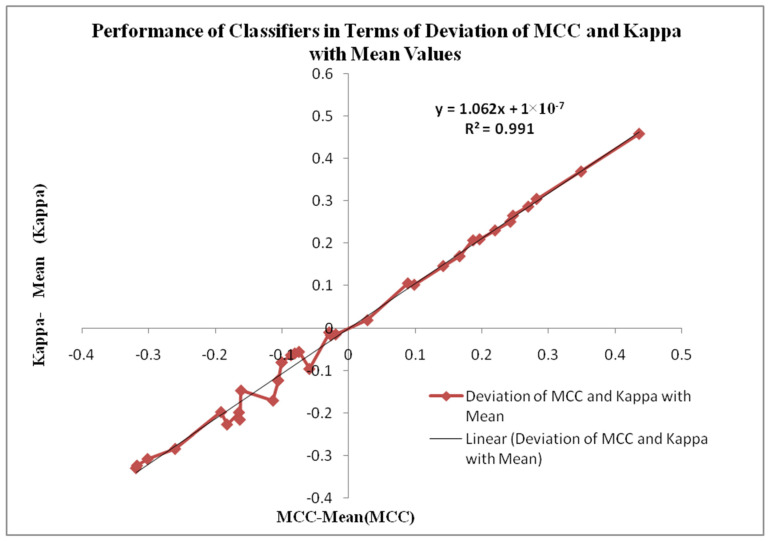
Performance of Classifiers in terms of Deviation of MCC and Kappa Parameters with mean Values.

**Table 1 bioengineering-10-00933-t001:** Average Statistical Features for STFT Dimensionally Reduced Adeno Carcinoma and Meso Cancer Cases.

Sl. No	Statistical Features	Cancer Classes	
		Adenocarcinoma	Meso Cancer
1	Mean	7527.993	9410.937
2	Variance	7,274,816	8,102,763
3	Skewness	14.7733	14.68975
4	Kurtosis	272.0948	270.2702
5	Pearson Correlation Coefficient (PCC)	0.993359	0.993084
6	f-test	0.011166	0.390122
7	*t*-test	0.264175	0.000004
8	*p*-value < 0.01	0.4958	0.98
9	Permutation Entropy	0.6931	0.6931
10	Sample Entropy	11.0007	11.0007
11	Canonical Correlation Analysis (CCA)	0.367333	

**Table 2 bioengineering-10-00933-t002:** Analysis of Friedman Test in Feature Selection Methods on STFT Data.

Sl. No	Feature Selection Methods	Parameters	Adeno Carcinoma Cancer	Meso Cancer
1	PSO	X^2^r statistic	2.326	1.376
		*p*-value	0.67604	0.84836
2	Harmonic Search	X^2^r statistic	6.39	1.18
		*p*-value	0.17185	0.88138

**Table 3 bioengineering-10-00933-t003:** Confusion matrix for Lung Cancer Detection.

Truth of Clinical Situation		Predicted Values	
		Adeno Carcinoma	Meso Cancer
Actual Values	Adeno Carcinoma	TP	FN
	Meso cancer	FP	TN

**Table 4 bioengineering-10-00933-t004:** Training and Testing MSE Analysis of Classifiers for STFT Dimensionality Reduction Technique without and with PSO and Harmonic Search Feature Selection.

Classifiers	Without Feature Selection		With PSO Feature Selection		With Harmonic Search Feature Selection	
	Training MSE	Testing MSE	Training MSE	Testing MSE	Training MSE	Testing MSE
Nonlinear Regression	1.68 × 10^−7^	5.63 × 10^−6^	4.9 × 10^−7^	1.32 × 10^−5^	6.25 × 10^−6^	9.6 × 10^−6^
GMM	3.6 × 10^−8^	7.21 × 10^−7^	6.76 × 10^−8^	4.84 × 10^−6^	9 × 10^−7^	1.96 × 10^−6^
SDC	3.25 × 10^−7^	8.84 × 10^−5^	2.89 × 10^−7^	3.61 × 10^−6^	2.35 × 10^−7^	1.16 × 10^−6^
Naïve Bayesian	5.48 × 10^−7^	6.24 × 10^−5^	1.09 × 10^−7^	1.52 × 10^−5^	1.32 × 10^−7^	6.76 × 10^−6^
SVM(Linear)	2.02 × 10^−7^	2.51 × 10^−5^	1.76 × 10^−7^	1.1 × 10^−6^	1.22 × 10^−7^	8.1 × 10^−5^
SVM(Poly)	9 × 10^−7^	3.48 × 10^−5^	4.9 × 10^−8^	3.03 × 10^−5^	6.4 × 10^−7^	7.06 × 10^−5^
SVM(RBF)	4.84 × 10^−7^	6.56 × 10^−5^	**2.25 × 10^−9^**	**3.6 × 10^−8^**	2.56 × 10^−8^	1.96 × 10^−7^

**Table 5 bioengineering-10-00933-t005:** Selection of Classifier Parameters.

Classifiers	Description
NLR	Class target T_1_ = 0.85 and T_2_ = 0.65, k_1_, k_2_, and k_3_ is from Equation (4), Convergence Criteria: MSE
GMM	Mean, Covariance of the input samples, and tuning parameters with test point likelihood probability 0.1, cluster probability of 0.5, with convergence rate of 0.6, Convergence Criteria: MSE
SDC	γ = 0.5, along with mean of each class target values as 0.65 and 0.85, Class Weights (w): 0.4, Bias (b): 0.05, Convergence Criteria: MSE
NBC	Smoothing parameter (alpha): 0.06, Prior Probabilities: 0.15, Convergence Criteria: MSE
SVM (Linear)	α (Regularization Parameter): 0.85, w: 0.4, b: 0.01, Convergence Criteria: MSE
SVM (Polynomial)	α: 0.76, Coefficient of the kernel function (γ): 10, w: 0.5, b: 0.01, Convergence Criteria: MSE
SVM (RBF)	α: 1, Kernel width parameter (σ): 100, w: 0.86, b: 0.01, Convergence Criteria: MSE

**Table 6 bioengineering-10-00933-t006:** Training and Testing Parameters of CNN Methodology for Raw Data and STFT Dimensionally reduced inputs.

Parameter	Value
Number of cross-validation	10
Learning rate	0.01
Number of mini-batch	100
Maximum number of epochs	10
Number of convolutional layer	3
Number of fully connected layer	4
Number of filter in first convolutional layer	16
Size of filter in first convolutional layer	3 × 3
Number of filter in second convolutional layer	32
Size of filter in second convolutional layer	3 × 3
Number of filter in third convolutional layer	64
Size of filter in third convolutional layer	3 × 3
Loss function	Cross-entropy

**Table 7 bioengineering-10-00933-t007:** Training and Testing Accuracy Analysis of various Classifiers in CNN Method with Raw Data and STFT features.

Classifiers	CNN Methods with Raw Data		CNN Methods with STFT Features	
	Training Accuracy	Testing Accuracy	Training Accuracy	Testing Accuracy
Nonlinear Regression	87.235 ± 1.64	83.42541 ± 2.41	89.327 ± 1.87	86.11111 ± 2.18
GMM	92.752 ± 1.76	89.50276 ± 2.74	91.458 ± 2.01	88.88889 ± 1.96
SDC	90.516 ± 2.42	87.8453 ± 2.25	94.372 ± 1.46	91.66667 ± 2.02
Naïve Bayesian	90.023 ± 1.53	86.74033 ± 2.68	**94.4527** ± 1.38	**91.66667** ± 2.45
SVM(Linear)	91.358 ± 1.72	87.8453 ± 1.74	91.753 ± 1.22	88.88889 ± 2.39
SVM(Poly)	90.145 ± 1.83	88.39779 ± 1.52	91.991 ± 1.94	88.88889 ± 1.51
SVM(RBF)	**93.631** ± 1.41	**90.60773** ± 1.32	92.047 ± 1.08	88.88889 ± 1.16

**Table 8 bioengineering-10-00933-t008:** Performance Analysis of Classifiers for STFT Dimensionality Reduction Technique without Feature Selection.

Classifiers	Parameters						
	Accuracy	Precision	Recall	F1 Score	MCC	Error Rate	Kappa
Nonlinear Regression	65.19337	93.069	62.666	74.9004	0.304098	34.80663	0.246382
GMM	80.66298	92.592	83.33	87.7193	0.441969	19.33702	0.428507
SDC	59.11602	90.42	56.666	69.67213	0.208379	40.88398	0.160987
Naïve Bayesian	60.77348	88.349	60.666	71.93676	0.167045	39.22652	0.137111
SVM(Linear)	72.37569	93.103	72.00	81.20301	0.362762	27.62431	0.321894
SVM(Poly)	71.8232	95.412	69.33	80.30888	0.409538	28.1768	0.348967
SVM(RBF)	64.64088	95.74	60.00	73.77049	0.355135	35.35912	0.274367

**Table 9 bioengineering-10-00933-t009:** Performance Analysis of Classifiers for STFT Dimensionality Reduction Technique with PSO Feature Selection.

Classifiers	Parameters						
	Accuracy	Precision	Recall	F1 Score	MCC	Error Rate	Kappa
Nonlinear Regression	56.90608	88.297	55.34	68.03279	0.149676	43.09392	0.115635
GMM	86.18785	96.296	86.66	91.22807	0.610382	13.81215	0.591791
SDC	81.76796	89.795	88.00	88.88889	0.382089	18.23204	0.381485
Naïve Bayesian	80.1105	96.29	86.67	86.95652	0.496771	19.8895	0.463968
SVM(Linear)	89.50276	95.172	92.00	93.55932	0.655197	10.49724	0.652451
SVM(Poly)	69.61326	90.59	70.67	79.40075	0.277252	30.38674	0.247373
SVM(RBF)	**94.47514**	**97.94**	**95.33**	**96.62162**	**0.817097**	**5.524862**	**0.814853**

**Table 10 bioengineering-10-00933-t010:** Performance Analysis of Classifiers for STFT Dimensionality Reduction Technique with Harmonic Search Feature Selection.

Classifiers	Parameters						
	Accuracy	Precision	Recall	F1 Score	MCC	Error Rate	Kappa
Nonlinear Regression	61.32597	94.44	56.66	70.83333	0.305452	38.67403	0.229319
GMM	88.95028	96.428	90.00	93.10345	0.664882	11.04972	0.654909
SDC	80.1105	93.1818	82.00	87.23404	0.44911	19.8895	0.430519
Naïve Bayesian	80.1105	90.14	85.33	87.67123	0.368107	19.8895	0.364417
SVM(Linear)	61.32597	93.47	57.34	71.07438	0.286204	38.67403	0.217998
SVM(Poly)	59.11602	88.00	58.67	70.4	0.151209	40.88398	0.121705
SVM(RBF)	**90.05525**	**97.82**	**90.00**	**93.75**	**0.711034**	**9.944751**	**0.696309**

**Table 11 bioengineering-10-00933-t011:** Performance Analysis of Classifiers for Raw Data Without Dimensionality Reduction with CNN Method.

Classifiers	Parameters						
	Accuracy	Precision	Recall	F1 Score	MCC	Error Rate	Kappa
Nonlinear Regression	83.42541	83.70	99.34	90.85366	0.170709	16.5745856	0.090147
GMM	89.50276	90.184	98.00	93.92971	0.583974	10.4972376	0.55643
SDC	87.8453	88.554	98.00	93.03797	0.501658	12.1546961	0.461601
Naïve Bayesian	86.74033	86.20	100	92.59259	0.441204	13.2596685	0.325885
SVM(Linear)	87.8453	88.095	98.67	93.08176	0.498308	12.1546961	0.443699
SVM(Poly)	88.39779	88.622	98.67	93.37539	0.52711	11.6022099	0.477669
SVM(RBF)	**90.60773**	**90.797**	**98.67**	**94.56869**	**0.632977**	**9.39226519**	**0.603121**

**Table 12 bioengineering-10-00933-t012:** Performance Analysis of Classifiers for STFT Dimensionality Reduction Technique with CNN Method.

Classifiers	Parameters						
	Accuracy	Precision	Recall	F1 Score	MCC	Error Rate	Kappa
Nonlinear Regression	86.11111	87.8787	96.67	92.06349	0.40452	13.88889	0.375
GMM	88.88889	90.625	96.67	93.54839	0.553399	11.11111	0.538462
SDC	**91.66667**	**93.54**	**96.67**	**95.08197**	**0.6825**	**8.333333**	**0.678571**
Naïve Bayesian	91.66667	90.909	100	95.2381	0.6742	8.333333	0.625
SVM(Linear)	88.88889	90.625	96.67	93.54839	0.553399	11.11111	0.538462
SVM(Poly)	88.88889	90.625	96.67	93.54839	0.553399	11.11111	0.538462
SVM(RBF)	88.88889	90.625	96.67	93.54839	0.553399	11.11111	0.538462

**Table 13 bioengineering-10-00933-t013:** Computational Complexity of the Classifiers for STFT Dimensionality Reduction Method without and with Feature selection methods and CNN Models.

Classifiers	Without Feature Selection	With PSO Feature Selection	With Harmonic Search	CNN with Raw Data	CNN STFT DR Method
Nonlinear Regression	O(2n^3^ log2n)	O(2n^6^ log2n)	O(2n^6^ log2n)	O(2n^4^ log2n)	O(2n^4^ log4n)
GMM	O(2n^4^ log2n)	O(2n^7^ log2n)	O(2n^7^ log2n)	O(2n^5^log2n)	O(2n^5^log4n)
Softmax Discriminant	O(2n^3^ log2n)	O(2n^6^ log2n)	O(2n^6^ log2n)	O(2n^4^ log2n)	O(2n^4^ log4n)
Naïve Bayesian	O(2n^3^ log2n)	O(2n^6^ log2n)	O(2n^6^ log2n)	O(2n^4^ log2n)	O(2n^4^ log4n)
SVM (Linear)	O(2n^3^ log2n)	O(2n^6^ log2n)	O(2n^6^ log2n)	O(2n^4^ log2n)	O(2n^4^ log4n)
SVM (Poly)	O(2n^4^ log2n)	O(2n^7^ log2n)	O(2n^7^ log2n)	O(2n^5^ log2n)	O(2n^5^ log4n)
SVM (RBF)	O(2n^2^ log4n)	O(2n^5^ log4n)	O(2n^5^ log4n)	O(2n^3^ log4n)	O(2n^3^ log8n)

**Table 14 bioengineering-10-00933-t014:** Comparison with Existing Works in Adenocarcinoma and Mesothelioma lung cancer classification from microarray gene datasets.

Serial No.	Author	Dataset	Methodology	Accuracy
1	As reported in this research work	LH2 Dataset	STFT with PSO feature selection and SVM (RBF) classifier	94.47
2	Gupta et al. (2022) [40]	Cancer Genome Atlas Dataset	Deep learning with CNN Methodology	92
3	Lin Ke et.al. (2022) [41]	LH2 Dataset	Decision Tree C4.5 Classification	93.2
4	Federica Morani et al. (2021) [42]	Cancer Genome Atlas and Curated Gene Expression Datasets	Multivariate approach using Cox	90
5	Fathi et al. (2021) [43]	LH2 Dataset	Feature combination from different network layers using Decision tree	85
6	Daniel Xia et al. (2020) [44]	LH2 Dataset	Minimalist approach for Classification	90.6
7	Azzavi (2015) et al. [45]	Kent Ridge Bio-medical Dataset	Multilayer perceptron (MLP), RBF and SVM	91.39 91.72 89.82
8	Guan et al. (2009) [46]	HG-U95 Dataset	SVM with RBF Kernel	94
9	Mramor et al. (2007) [47]	Whitehead Institute Database	SVM, NBC, k-nearest neighbor (KNN), and Decision Tree	94.67 90.35 75.28 91.21
10	Gordon (2002) [21]	LH2 Dataset	Translation of microarray data to gene expressions	90

**Table 15 bioengineering-10-00933-t015:** Comparison of previous works involving lung and other types of cancer classification from microarray gene datasets.

Serial No.	Type of Cancer and Dataset	Methodology	Accuracy	Precision	Recall	F1 Score
1	Lung cancer—LH2 Dataset [21]	GMM Classifier with STFT DR and without feature selection	80.66	92.592	83.33	87.7193
		SVM (RBF) Classifier with STFT DR and PSO feature selection	90.475	97.94	95.33	96.62162
		SVM (RBF) Classifier with STFT DR and HS feature selection	90.055	97.82	90	93.75
		SVM (RBF) Classifier with CNN and Raw Data	90.607	90.797	98.67	94.56869
		Softmax Discriminant Classifier with CNN and STFT DR	91.666	93.54	96.67	95.08197
2	Lung cancer—Gordon (2002) [21]	PCC-DTCV model with optimized DT and PPC ≥ 0.4	93	84	84	84
		PCC-DTCV model with optimized DT and PPC ≥ 0.5	94	77	77	77
		PCC-DTCV model with optimized DT and PPC ≥ 0.6	95	92.4	77	84
3	Prostate cancer—Singh (2002) [49]	SVM classifier with Elastic Net	95.9936	96.4286	96.131	96.27957
		SVM classifier with Elastic Net CV	99.4565	96.4286	96.131	96.27957
		MLP classifier with Lasso	95.1122	96.4286	96.131	96.27957
		MLP classifier with Elastic Net	96.0737	98.2143	98.2143	98.2143
4	Breast cancer—Chin (2006) [50]	SVM classifier with Elastic Net	86.6071	87.3106	72.5	79.2190067
		SVM classifier with Elastic Net CV	89.881	87.3106	95.9722	91.4367345
		MLP classifier with Lasso	87.4405	79.1667	89.3687	83.9589198
		MLP classifier with Elastic Net	88.9881	89.5076	79.1667	84.0201657
5	Leukemia—Golub (1999) [51]	SVM classifier with Elastic Net	97.2222	96.875	97.9167	97.3930646
		SVM classifier with Elastic Net CV	97.2222	96.875	97.9167	97.3930646
		MLP classifier with Lasso	95.8333	100	100	100
		MLP classifier with Elastic Net	97.2222	95.8333	93.75	94.7802035

## Data Availability

Not applicable.

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
