# Peer review of "Evaluation and Exploration of Machine Learning and Convolutional Neural Network Classifiers in Detection of Lung Cancer from Microarray Gene—A Paradigm Shift"

_bioengineering, 2023, doi:10.3390/bioengineering10080933_

Round 1

Reviewer 1 Report

This manuscript investigates the use of a two-level strategy involving feature extraction and selection methods before the classification step. The manuscript is extensive and the authors should be commended for this. However, there the figures, tables, equations require amending to be clearer to the reader. The results of the SVM in Table 12 are concerning, requiring clarification as to why they are the same. Detailed comments are below:

-          Quantify the class imbalance for Adenocarcinoma and malignant Mesothelioma samples

-          Make sure all your equations have defined all characters. For example, Equation (1) – define M, j and w; Equation 4 – define j; Equation 7 = define g.

-          Place Table 1 after you introduce it in your main text

-          Ensure all acronyms are defined in your text at the first occurrence (e.g. CCA)

-          Figure 2 – legend covers scatter and place figure after first mentioned in text. More discussion for the reader about this plot please.

-          Please check your manuscript as some text format errors (e.g. “themicroarray, “cancer.Since”, “classes.Shah”)

-          Figures 3 & 4– please provide a legend for the colours and what the divider line represents

-          Figure 5 – please provide a legend for the colours

-          Table 2 – p<.01 redundant as p-value provides significance

-          Table 4 – Please provide std error for your MSE

-          Table 5 is not explained in your manuscript text.

-          Please explain the parameter tuning performed for all your models that resulted in those chosen in Table 5

-          Table 7 – Please clarify why the Testing Accuracy was exactly the same for many of the models for the STFT Features

-          It is unclear why you have 4.2. CNN with cross entropy loss section just prior to the Discussion. Perhaps this section should be within the original CNN methods explanation.

-          Your explanation about accuracy is after you have already presented accuracy metrics which is confusing

-          Please include precision and recall in your results metrics

-          Please include acceptable ranges for the Kappa

-          Figure 9 – it is unclear what the y-axis represents – please clarify

-          Figure 10 – it is unclear what the percentage represents on the y-axis in your figure – please clarify. Also, your legend does not include all metrics/colours (e.g. lavender, pink)

-          Table 11 – please define colours under your table.

-          Table 12 – The metrics for SVM are exactly the same which does not make sense. Please clarify the reason for this.

-          Table 14 – please include more than accuracy as a comparator as this can sometimes be misleading, especially for imbalanced target

There are numerous occasions where words are not separated. Please thoroughly check the manuscript for English and format errors.

Author Response

Dear Reviewer,

Thank you for recommending a major revision to the manuscript and allowing us a resubmission of our manuscript. We have responded to each of the comments accordingly.

Best Regards,

Authors

Reviewer 2 Report

Authors tested several machine-learning and deep-learning-based classifiers on the microarray gene expression data for lung cancer. They analyzed a two-level classification approach, in which first some informative features extracted from the data (dimensionality reduction followed by feature selection) and then the classification is performed on the selected features. Based on the results for the previously published microarray data, they reported best specific feature extraction and selection methods and a best classifier. The study is very detailed and contains the connection with previous results.

I have the following questions/remarks:

1.     Title: I’m not sure the term ‘paradigm shift’ is relevant, since the data preprocessing (feature extraction) used by the authors is not something new in the classification problems on the biomedical data. If the ‘shift’ is not meant to be related with this, please explain.

2.     Abstract, last sentence: Information about data related to the reported performance should be added.

3.     p.6: “Table 1 shows the statistical features analysis for three types of Dimensionally Reduced Adeno carcinoma and Meso Cancer Cases of microarray genes.” – What are the ‘three types’? I can see only two types of cancer.

4.     Table 1: It’s not quite clear which characteristics were correlated with each other within each cancer class – it’s just the extracted features? If so, it’s not clear precisely what statistical hypotheses were tested with the F- and t-tests (lines 6 and 7 of the table) within each cancer class? For t-test, the statistic has quite a small value (line 7 of the table) -- is it still not statistically significant?

5.     Figures 3--5: What does color coding mean in the figures?

6.     Table 2: Please specify the statistical hypotheses for p-values in the table.

7.     p.16: “the targets … are chosen as 0.95, and 0.1, respectively.” -- Why these targets are not equidistant from 1 and 0, respectively?

8.     Table 4 and 7: Does ‘testing’ here in fact mean the cross-validation estimates? If so, it would be more informative to see mean ± error values from k-fold cross-validation. You could even estimate the statistical significance for the difference between methods using those performance values (here and in similar tables further in the text).

9.     Table 8--12: Are these the cross-validation estimates (in this case, mean ± error would be more informative) or the performance on the testing datasets?

Author Response

(The authors gave the same response as above.)

Round 2

Reviewer 1 Report

Thank you for addressing my comments and suggestions. Table 7, with the inclusion of the std erros, do indicates there is much overlap with the accuracy so your evaluation of this table requires updating. For example,  "In the STFT feature method, Naïve 725 Bayesian Classifier arrived at maximum training accuracy of 94.452%." but the std error is +/- 1.38 so this accuracy overlaps other accuracies. Please adjust your manuscript accordingly.

Author Response

Dear Reviewer,

Thank you for recommending a minor revision to the manuscript and allowing us to submit a revision of our manuscript, with an opportunity to address the reviewer comment. We have responded to the reviewer comment accordingly.

Best regards,

Authors
